# Emergence of representative signals for sudden stratospheric warmings beyond current predictable lead times

Zheng Wu[1], Bernat Jiménez-Esteve[1], Raphaël de Fondeville[2], Enikő Székely[2], Guillaume Obozinski[2], William T. Ball[3], and Daniela I.V. Domeisen[1]

[1]Institute for Atmospheric and Climate Science, ETH Zürich, Switzerland
[2]Swiss Data Science Center, ETH Zürich and EPFL, Switzerland
[3]Department of Geoscience and Remote Sensing, Faculty of Civil Engineering and Geosciences, TU Delft, the Netherlands

**Correspondence:** Zheng Wu (zheng.wu@env.ethz.ch)

**Abstract.** Major sudden stratospheric warmings (SSWs) are extreme wintertime circulation events of the Arctic stratosphere that are accompanied by a breakdown of the polar vortex and are considered an important source of predictability of tropospheric weather on subseasonal to seasonal time scales over the Northern Hemisphere mid- and high- latitudes. However, SSWs themselves are difficult to predict, with a predictability limit of around one to two weeks. The predictability limit for determining the type of event, i.e., wave-1 or wave-2 events, is even shorter. Here we analyze the dynamics of the vortex breakdown and look for early signs of the vortex deceleration process at lead times beyond the current predictability limit of SSWs. To this end, we employ a mode decomposition analysis to study the potential vorticity (PV) equation on the 850 K isentropic surface by decomposing each term in the PV equation using the empirical orthogonal functions of the PV. The first principal component (PC) is an indicator of the strength of the polar vortex and starts to increase from around 25 days before the onset of SSWs, indicating a deceleration of the polar vortex. A budget analysis based on the mode decomposition is then used to characterize the contribution of the linear and nonlinear PV advection terms to the rate of change (tendency) of the first PC. The linear PV advection term is the main contributor to the PC tendency at 25 to 15 days before the onset of SSW events for both wave-1 and wave-2 events. The nonlinear PV advection term becomes important between 15 to 1 days before the onset of wave-2 events, while the linear PV advection term continues to be the main contributor for wave-1 events. By linking the PV advection to the PV flux, we find that the linear PV flux is important for both types of SSWs from 25 to 15 days prior to the events but with different wave-2 spatial patterns, while the nonlinear PV flux displays a wave-3 wave pattern, which finally leads to a split of the polar vortex. Early signs of SSW events arise before the one to two week prediction limit currently observed in state-of-the-art prediction systems, while signs for the type of event arise at least one week before the event onset.

## 1 Introduction

Major sudden stratospheric warmings (SSWs) (Baldwin et al., 2021) are extreme wintertime circulation events of the Arctic stratosphere that are accompanied by a breakdown of the polar vortex which consists of strong circumpolar westerly winds

in the polar stratosphere that form in fall and decay in spring. During a major SSW event the zonal-mean zonal wind in the stratosphere reverses in mid-winter from westerly to easterly, accompanied by an abrupt increase of temperatures in the entire polar stratosphere (Labitzke, 1981). SSWs are caused by the interaction between planetary waves and the mean flow in the stratosphere (Matsuno, 1971; McIntyre, 1982). The planetary waves are generated in the troposphere by flow over mountains, land-sea thermal contrast, and nonlinear synoptic scale wave-wave interactions (Charney and Eliassen, 1949; Scinocca and Haynes, 1998; Held et al., 2002; Domeisen and Plumb, 2012). The waves can propagate upward into the stratosphere if their wave number is sufficiently small (e.g., wave-1 and wave-2 components) and if the background zonal-mean zonal wind is eastward relative to the zonal phase speed of the waves (Charney and Drazin, 1961). When these planetary waves reach a critical level in the stratosphere, they break and deposit easterly momentum into the mean flow, resulting in a deceleration of the mean flow, which can eventually lead to a breakdown of the polar vortex, and an SSW event if the winds reverse to easterlies (Charlton and Polvani, 2007). The predictability limit for SSW events in state-of-the-art subseasonal prediction systems is around one to two weeks (Domeisen et al., 2020b). After an SSW event, the stratospheric anomalies can propagate downward to the lower stratosphere and influence the tropospheric weather for up to two months after the onset of events (Baldwin and Dunkerton, 2001; Kidston et al., 2015). For example, SSWs are found to be associated with an anomalously negative phase of the North Atlantic Oscillation (Domeisen, 2019) and an equatorward shift of the tropospheric extratropical jet streams (Baldwin and Dunkerton, 2001; Limpasuvan et al., 2004). The shift of the jet is crucial for the weather over North America and Europe, as it can lead to a larger probability of cold air outbreaks (Kolstad et al., 2010; King et al., 2019). Therefore, SSWs are thought to be an important source of predictability on subseasonal to seasonal (S2S) time scales over the Northern Hemisphere (NH) mid- and high latitudes (Mukougawa et al., 2009; Scaife et al., 2016; Karpechko et al., 2017). Improving the predictability of SSW events may therefore help to enhance the forecast skill in the troposphere (Sigmond et al., 2013; Domeisen et al., 2020a).

Even though the polar vortex undergoes deceleration and disruption during all major SSW events, there are large differences amongst SSW events in terms of their dynamical evolution, vortex structure, and downward impact on the troposphere. Based on the geometry of the polar vortex at the onset of the event, SSWs can be classified into two types: 1) vortex displacement events, when the vortex is shifted off the pole, and 2) vortex split events, when the vortex is split into two parts (Charlton and Polvani, 2007). While displacement events are mainly attributed to the enhanced upward propagation of wavenumber 1 waves (hereafter: wave-1), split events are often related to strong wavenumber 2 waves (hereafter: wave-2) (Nakagawa and Yamazaki, 2006). In observations, major SSWs occur in about two out of three winters (Charlton and Polvani, 2007) with a similar frequency of split and displacement events (Butler et al., 2015), but with high decadal variability (Domeisen, 2019). However, if SSWs are classified based on the zonal wavenumber of the wave flux in the lower stratosphere, there are more wave-1 events than wave-2 events as not all split SSWs are dominated by wave-2 wave flux (Bancalá et al., 2012; Ayarzagüena et al., 2019). The occurrence of split events tends to be less predictable than that of displacement events, especially at lead times of 1-2 weeks (Taguchi, 2018; Domeisen et al., 2020b). Given the fact that the development of the two types of SSW events is considered to be different (Matthewman et al., 2009; Albers and Birner, 2014), the dynamical processes that lead to

the breakdown of the polar vortex should be distinct between displacement (wave-1) and split (wave-2) events and should also be distinguishable from normal winter days (without SSWs). Therefore, understanding the dynamics of the vortex disruption and identifying signals that contribute to the vortex deceleration are crucial for improving the predictability of SSWs and of each type of event, and ultimately, of the weather at the Earth's surface.

Since the stratospheric circulation is well described by Ertel's potential vorticity (PV) (McIntyre, 1982), the evolution of the polar vortex during SSWs can also be captured by the changes in the values and structure of the PV in the stratosphere. As discussed above, while the polar vortex undergoes breakdown in each major SSW event, the associated vortex structures are different for the two types of SSW events. Decomposing the PV into an empirical orthogonal function (EOF) basis, we can identify the PV structure that best describes the weakening of the polar vortex and subsequently investigate how its correspond-
ing principal component (PC) changes with time. By projecting the other variables from the PV equation (i.e., the zonal and meridional wind) onto the EOF basis from PV, one can analyze the contribution of each term of the equation to the changes in the PC time series in order to identify the dynamical processes that are the most relevant for the weakening of the polar vortex. This approach was proposed by Aikawa et al. (2019) and called *mode decomposition analysis*. Aikawa et al. (2019) applied the mode decomposition analysis to diagnose the atmospheric blocking development in the Eastern Pacific and Central Atlantic,
and demonstrated that the blocking index can be faithfully reconstructed using only the first 10 EOF modes. The vorticity equation was then decomposed into three terms (i.e., linear advection, nonlinear mode-to-mode interaction, and dissipation), and their contribution to the combined time evolution of the first 10 PC time series was subsequently investigated. Their results showed that the nonlinear interaction terms contribute to the increase in the amplitude of the blocking index in both regions (Eastern Pacific and Central Atlantic), but that their contributions are different. Since each term in the vorticity equation can be
linearly reconstructed using the EOF modes (that correspond to specific spatial patterns) and the PCs, this method allowed the identification of the wind and vorticity patterns that are crucial for the development of the blocking. As the results from Aikawa et al. (2019) indicate the effectiveness of mode decomposition analysis in studying dynamically-driven events, we use the same method to study the development of SSWs, which are also driven by wave dynamics (e.g., Matsuno, 1971). Indeed, we find that the vortex weakening can be represented by the evolution of the EOF modes extracted from PV. We therefore employ a budget
analysis of the PV equation in the stratosphere to quantify the contribution of each EOF mode to the dynamical processes that lead to the deceleration of the polar vortex and the subsequent onset of SSWs.

The onset of SSW events is associated not only with the anomalously large excitation of wave activity in the troposphere (Matsuno, 1971; Polvani and Waugh, 2004; Lindgren et al., 2018), but also with the stratospheric mean state and stratospheric wave anomalies prior to SSWs (Hitchcock and Haynes, 2016; Jucker, 2016; Birner and Albers, 2017; de la Cámara et al.,
2019). Moreover, split and displacement SSW events exhibit distinct pre-warming evolutions (Charlton and Polvani, 2007; Matthewman et al., 2009; Bancalá et al., 2012; Albers and Birner, 2014). For example, the zonal wavenumber of the wave flux leading to the breakdown of the polar vortex can be different for the two types of SSW events (e.g., Bancalá et al., 2012). Some studies suggest that the explosive growth of wave amplitude is triggered by resonant behavior, which is also different between the two types of SSW events (e.g., Esler and Matthewman, 2011; Matthewman and Esler, 2011). Albers and Birner

(2014) further suggest that different effects of planetary Rossby and/or gravity waves are responsible for producing the distinct vortex preconditioning that is conducive to developing the respective split and displacement SSWs events. Given the distinct dynamical developments of the two types of SSWs, one should be able to observe different evolutions of the PV terms by the mode decomposition analysis for each type of SSW event. Since displacement (split) events are mainly related to wave-1 (wave-2) wave activity, in this study, SSWs are classified into wave-1 and wave-2 events based on the dominant wavenumber

that leads to the breakdown of the vortex. As some studies point out that the dynamical process of the vortex breakdown starts earlier than the current predictable lead time (meaning the lead time on which an event can be predicted) of two weeks (Polvani and Waugh, 2004; Jucker and Reichler, 2018), one would expect to see signals indicative of the vortex breakdown appearing in the mode equation budget before the onset of the SSWs. Our goal in the current study is therefore to identify signals that are representative of SSWs, i.e., distinct from normal winter days, ahead of the vortex breakdown with lead times longer than two

100  weeks, and to distinguish onset signals for wave-1 and wave-2 events beyond the currently achieved predictable lead times.

The paper is organized as follows. Section 2 describes the data used in the analyses and the methodology behind the mode decomposition analysis. Section 3 shows the results of the analysis and their implications. Section 4 further provides the physical interpretation of the signals found in the mode equation budget by linking them with wave-mean flow interactions. Conclusions are given in Section 5.

## 2   Data and methodology

### 2.1   Data and EOF basis

We use two data sets for analysis in this paper: 1) the ERA-Interim reanalysis (Dee et al., 2011), and 2) simulations from an intermediate complexity configuration of the Isca model (Vallis et al., 2018) (hereafter the Isca model). This version of the Isca model uses the model configuration from Jiménez-Esteve and Domeisen (2019), and it uses a T42 horizontal resolution

and 50 vertical levels up to 0.02 hPa with 25 levels above 200 hPa. The model includes moist and radiative processes through evaporation from the surface and fast condensation. Water vapour in the atmosphere interacts with a multiband radiation scheme (Mlawer et al., 1997) and a simple Betts-Miller convection scheme (Betts and Miller, 1986). The $CO_2$ concentration is fixed at 300ppm and the seasonal cycle of ozone in the stratosphere is prescribed based on the ERA-Interim (1979-2016) climatology. For the lower boundary conditions, the Isca model uses realistic topography and the continental outline from the ECMWF

model, and sea surface temperatures (SST) are prescribed. The model does not include a representation of clouds, interactive chemistry, or gravity wave drag. In this paper, we use the experiment that uses prescribed strong El Niño-like SST anomalies as described in Jiménez-Esteve and Domeisen (2019). The motivation to use this model experiment is that it produces a realistic climatology and a frequency of SSWs that is similar to the reanalysis.

For ERA-Interim, we use daily mean fields of potential vorticity (PV) $P$, zonal wind $u$, and meridional wind $v$ at the 850 K isentropic level from 1979-2018 with a horizontal resolution of $2.5° \times 2.5°$. Only data north of 30°N (24 latitude values by 144 longitude values leading to a total of $D = 3456$ grid points) from the winter season (October to April, 8490 days in total) are included in the analysis. For the Isca model simulations, the daily vertical gradient of potential temperature ($\theta$), zonal wind $u$, and meridional wind $v$ are interpolated to the 850 K isentropic surface from pressure levels. The Isca model data contains a total of 130 years, corresponding to 27300 winter days. The PV is computed as

$$P = (\zeta_\theta + f)\left(-g\frac{\partial \theta}{\partial p}\right), \tag{1}$$

where $P$ is Rossby-Ertel's PV (Hoskins et al., 1985), $\zeta_\theta$ is the relative vorticity on the isentropic surface, $\theta$ is the potential temperature, $f$ is the Coriolis force, $g$ is the gravity, and $p$ is the pressure. Note that in this work potential vorticity (PV) refers to Rossby-Ertel's PV.

The PV data (for both the ERA-Interim reanalysis and the Isca model) is further decomposed into: 1) the daily climatology, obtained by computing the daily mean values of PV over all available years, and 2) daily anomalies with respect to the climatological seasonal cycle. Given the limited number of years in the reanalysis, we also applied a 30-day running mean to the daily climatology to remove the high-frequency variability and repeat the analysis performed in the study. The results are almost identical to the ones using climatologies without low-pass filtering, which is mainly due to the fact that applying PCA acts as a filtering and indicates the importance of the low-frequency components in the evolution of the polar vortex. In the following figures, we show the results using climatologies without low-pass filtering. The EOF modes of the PV and their corresponding PC time series (later used in the mode decomposition analysis) are obtained by employing principal component analysis (PCA) on the daily PV anomalies at 850 K. The procedure consists of applying PCA twice, as described in the following. We apply a first PCA only to the PV data around the onset date of all SSWs (from -10 to +5 days around the onset date). The motivation behind this first PCA is to obtain a mode that captures the characteristics of SSWs. The spatial pattern of the resulting first EOF mode ($E_1 \in \mathbb{R}^D$) is shown in the first panel in Figure 1, with a wavenumber 0 structure centered at the pole. Next, we project the whole winter data (October to April) onto the subspace orthogonal to $E_1$ by subtracting from the winter data its projection onto $E_1$, and then perform a second PCA on this projection. The resulting data does not contain any information from $E_1$ since it is in the space orthogonal to $E_1$, and yields a total of $D = 3456$ modes $\{E_2, \ldots, E_{D+1}\}$ (equivalent to the number of grid points). Combining the first EOF mode $E_1$ from the first PCA with the $D$ EOF modes from the second PCA forms an orthogonal basis for the whole winter data, referred to in the following as the "combined set of basis vectors". Figure 1 shows the spatial pattern of the first 10 EOF modes that explain together $\approx 71\%$ of the variance of the PV anomalies of all winter days in the ERA-Interim data. The spatial patterns of the first 10 EOF modes of the PV anomalies in the Isca model data is shown in Figure C1 in Appendix C. We note that the EOF modes cannot be interpreted by default as physical modes as discussed also in Dommenget and Latif (2002) and Monahan et al. (2009). Here, since $E_1$ is derived using only days around SSWs (-10 to +5 days around the onset date of SSWs) and displays a clear zonally symmetric structure, the physical process represented by the variation of $E_1$ can be interpreted as representing the changes in the strength of the polar vortex during SSWs. In ERA-Interim, the variance explained by $E_1$ (15.8%) for the whole winter data is slightly smaller than

that of $E_2$ (16.8%), as shown in Figure 1. The EOF modes are orthogonal and therefore satisfy the following relationship of the inner product $\langle \cdot, \cdot \rangle$,

$$155 \quad \langle E_m, E_n \rangle = \int_\Omega E_m E_n \, dx = \delta_{mn}, \quad m, n = 1, \ldots, D+1, \tag{2}$$

where $\delta_{mn}$ is the Kronecker delta function, and $\Omega$ is the region poleward of 30°N. As mentioned above, the main motivation for applying the double PCA approach to obtain the EOF basis rather than directly applying PCA once to all the winter days is that $E_1$ is computed from the days around the onset day of SSWs, and its variability is therefore closely linked to SSW events, as can be seen from Figure 2. The PC time series of all winter days associated with $E_1$ (Figure 2) is highly correlated with the

polar-cap averaged temperature at 30 hPa, thus being a good representation of SSWs (Blume et al., 2012). The evolution of the first PC (for all winter data) therefore enables us to better understand the vortex breakdown process.

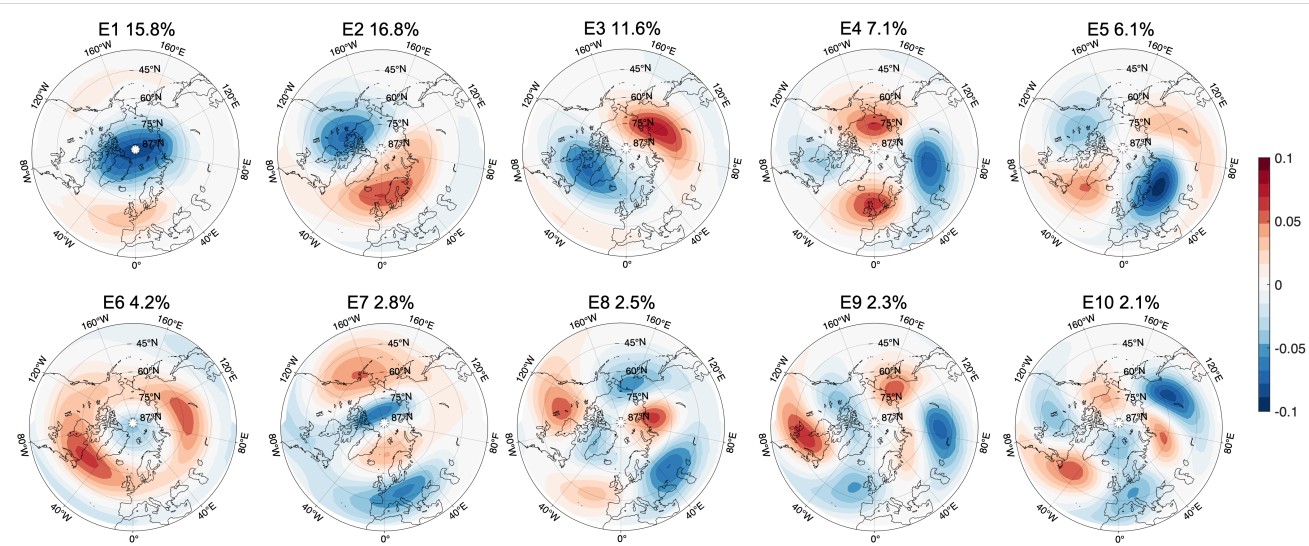

**Figure 1.** The first 10 EOF spatial patterns of PV at 850 $K$ ($E_1, E_2, \ldots, E_{10}$) of the combined set of basis vectors as described in the text using ERA-Interim daily data. The percentage number indicates the variance explained by each EOF.

## 2.2    Mode decomposition analysis

Following the methodology from Aikawa et al. (2019), the mode decomposition analysis is applied to the PV conservation equation to study the dynamical development of SSW events. The anomalous PV equation on an isentropic surface can be

written by separating the daily anomalies from the daily climatological mean for 1979-2018 as

$$\frac{\partial P_a}{\partial t} + \boldsymbol{V}_c \cdot \nabla P_a + \boldsymbol{V}_a \cdot \nabla P_c + \boldsymbol{V}_a \cdot \nabla P_a = F_a, \tag{3}$$

where $P$ is the PV, $\boldsymbol{V}$ is the wind vector field, and $F$ is the forcing term (computed here as residual). The subscript $c$ refers to daily climatology, and $a$ refers to daily anomalies. The anomalous PV tendency $\left(\frac{\partial P_a}{\partial t}\right)$ equals the sum of linear effects that consist of the advection of daily anomalous PV by climatological wind vector ($\boldsymbol{V}_c \cdot \nabla P_a$) and the advection of climatological PV by the anomalous wind vector ($\boldsymbol{V}_a \cdot \nabla P_c$), and the nonlinear effects of the advection of anomalous PV by the anomalous wind vector ($\boldsymbol{V}_a \cdot \nabla P_a$).

For the mode decomposition analysis of the PV equation, we focus only on the time frame between 50 to 1 days prior to the onset day of SSW events as the goal of this study is to identify signals that precede SSWs. We project the PV field (obtained by concatenating the data from -50 to -1 days before the onset of all SSWs) onto the first $d = 1000$ modes of the combined set of basis vectors $\{E_1, E_2, \ldots, E_d\}$ and get the corresponding PC time series, denoted as $\{A_1, A_2, \ldots, A_d\}$. We found that the first $d = 1000$ modes (which in total explain 99.9% of the variance of PV as shown in Figure A1a in Appendix A) are sufficient to reproduce the actual rate of change of the PCs. The PV daily anomalies ($P_a$) can be expressed as the linear combination of $\{E_1, E_2, \ldots, E_d\}$ with coefficients $\{A_1, A_2, \ldots, A_d\}$. For the wind vector daily anomalies ($\boldsymbol{V}_a$), the temporal evolution of PV is correlated with that of the wind fields. Therefore, it is possible to obtain a set of spatial patterns $\{\boldsymbol{U}_1, \boldsymbol{U}_2, \ldots, \boldsymbol{U}_d\}$ for $\boldsymbol{V}_a$ by regressing $\boldsymbol{V}_a$ onto the PC time series of PV, $\{A_1, A_2, \ldots, A_d\}$. Note that the spatial patterns $\{\boldsymbol{U}_1, \boldsymbol{U}_2, \ldots, \boldsymbol{U}_d\}$ are not orthogonal as they are not obtained through an EOF mode decomposition. The $\boldsymbol{V}_a$ can then be represented as the linear combination of $\boldsymbol{U}_1, \boldsymbol{U}_2, \ldots, \boldsymbol{U}_d$ with coefficients $A_1, A_2, \ldots, A_d$. Then, by substituting the projection of $P_a$ and $\boldsymbol{V}_a$ onto the PC time series into Eq. (3) and taking the inner product between Eq. (3) and a given EOF mode $E_k$, we obtain the mode equation budget of the rate of change (or tendency) of $A_k$ as (detailed derivation in Appendix A)

$$\frac{dA_k}{dt} = \frac{1}{C_k}\left(-\sum_{n=1}^{d} L_{kn}^A A_n - \sum_{n=1}^{d} L_{kn}^B A_n - \sum_{m=1}^{d}\sum_{n=1}^{d} N_{kmn} A_m A_n + F_k\right), \tag{4}$$

where $C_k = \langle A_k, A_k\rangle$ is the eigenvalue associated with mode $E_k$; $L_{kn}^A = \langle E_k, \boldsymbol{V}_c \cdot \nabla E_n\rangle$ and $L_{kn}^B = \langle E_k, \boldsymbol{U}_n \cdot \nabla P_c\rangle$ are the inner products between the linear advection terms ($\boldsymbol{V}_c \cdot \nabla E_n$ and $\boldsymbol{U}_n \cdot \nabla P_c$) and $E_k$; $N_{kmn} = \langle E_k, \boldsymbol{U}_m \cdot \nabla E_n\rangle$ is the inner product between the nonlinear advection term ($\boldsymbol{U}_m \cdot \nabla E_n$) and $E_k$; and $F_k = \langle E_k, F_a\rangle$ is the residual term. The sum of the two linear advection terms gives the total linear advection term. Using Eq. (4), we then compute the contribution of each mode to the linear and nonlinear advection terms and thus to the total rate of change of $A_k$ to determine which modes (or combinations of modes) play an important role in identifying signals distinguishing SSWs from normal winter days.

From the power spectrum of each PC time series $A_k$ (see Figure A1b in Appendix A), the power of $A_1$ to $A_{25}$ is concentrated at frequencies lower than once a week, which is different from the power spectrum of the other PCs beyond $A_{25}$. Based on these power spectra, here we consider the associated modes $E_1$ to $E_{25}$ as low modes, which together explain around 85% of the total variance, and modes beyond $E_{25}$ as high modes. To separate the contributions from low and high EOF modes, the summation over all modes from Eq. (4) is divided into the summation of low modes and that of high modes. Thus, Eq. (4) can

be written as

$$\frac{dA_k}{dt} = \frac{1}{C_k}\left(-\sum_{n=1}^{l}(L_{kn}^A + L_{kn}^B)A_n - \sum_{n=l+1}^{d}(L_{kn}^A + L_{kn}^B)A_n\right.$$

$$\left. - \sum_{m=1}^{l}\sum_{n=1}^{l}N_{kmn}A_mA_n - \sum_{m=1}^{l}\sum_{n=l+1}^{d}(N_{kmn} + N_{knm})A_mA_n - \sum_{m=l+1}^{d}\sum_{n=l+1}^{d}N_{kmn}A_mA_n + F_k\right)$$

$$= \frac{1}{C_k}(L_{low} + L_{high} + N_{low-low} + N_{low-high} + N_{high-high} + F_k)$$

$$= \frac{1}{C_k}(L_{total} + N_{total} + F_k), \tag{5}$$

where $l = 25$ represents the $n$th mode that is treated as the last low mode; $L_{low}$ and $L_{high}$ are the linear low and high mode terms, respectively; $N_{low-low}$, $N_{low-high}$ and $N_{high-high}$ are the contributions from nonlinear low-low, low-high, and high-high mode interactions, respectively; $L_{total}$ is the total linear PV advection as $L_{total} = L_{low} + L_{high}$ and $N_{total}$ is the total nonlinear PV advection as $N_{total} = N_{low-low} + N_{low-high} + N_{high-high}$. We performed sensitivity tests for a range of $l$ values, and our results and conclusions are robust for values of $l > 25$. Moreover, the contribution from modes higher than $l = 25$ is very small (not shown). In this study, we only focus on the rate of change of the first PC time series $\frac{dA_1}{dt}(k=1)$ due to the strong relationship between $A_1$ and SSWs.

### 2.3 SSW definition

The major SSW definition follows the criterion of Charlton and Polvani (2007), based on the reversal of the daily zonal-mean zonal winds at 10 hPa and 60°N, and a return to westerlies afterward for at least 10 consecutive days before the final warming. Two events in the same season are treated as distinct SSWs if they are separated by at least 20 days. The central dates of split and displacement SSW events in ERA-Interim before 2014 are taken from Karpechko et al. (2017) and consist of 11 split events and 12 displacement events. In addition, we added the SSW event in 2018, which is classified as split event based on the vortex geometry (Charlton and Polvani, 2007). The definition of wave-1 and wave-2 SSW events is based on the eddy heat flux at 100 hPa and 60°N similar to Bancalá et al. (2012). If the wave-2 component of eddy heat flux is larger than the wave-1 component by 15 $\mathrm{Kms}^{-1}$ in the period of -2 to 0 days before the SSW event for at least 1 day, then the SSW event is classified as a wave-2 event, otherwise as a wave-1 event. Note that the time window around the SSW event (day -2 to day 0) that is used to classify the type of event is shorter than that in Bancalá et al. (2012). The reason for using a shorter window is to reduce the overlap between the time interval used to define the type of SSW event and the lead times that emerge as relevant in the predictability of wave-1 vs. wave-2 events. According to this definition, there are 18 wave-1 and 7 wave-2 SSW events in ERA-Interim. Note that not all split events are dominated by wave-2 wave flux (only 6 out of 12 split SSWs are also classified as wave-2 events), while one displacement event is dominated by wave-2 wave flux according to this definition.

## 2.4 Interpretation of results from mode decomposition based on PV flux

Up to this point, our mode decomposition analysis does not employ any explicit approximation. In this section, we will demonstrate how $L_{total}$ and $N_{total}$ are related to the poleward PV flux, which can help to illustrate the physical interpretation of the results obtained from the mode decomposition analysis.

In order to provide a physical interpretation for each term in Eq. (5) and understand the physical process that can lead to the disruption of the polar vortex, we introduce the poleward flux of PV on an isentropic surface given by Tung (1986) as

$$\nabla \cdot \mathcal{F} = [\rho_\theta][(\rho_\theta P^{**})v^*]\cos\phi, \tag{6}$$

where $\mathcal{F}$ is the Eliassen-Palm flux (EP flux), $P$ is the potential vorticity, $v$ is the meridional wind, $\rho_\theta$ is the density in isentropic coordinates, defined as $\rho_\theta = -\frac{1}{g}\frac{\partial p}{\partial \theta}$, and $\phi$ is the latitude. The brackets denote the zonal mean, one asterisk denotes the deviation from the zonal mean, and two asterisks denote the deviation from the density-weighted zonal average, as in

$$P^{**} = P - \frac{[\rho_\theta P]}{[\rho_\theta]}. \tag{7}$$

The formulation of the PV flux in Eq. (6) is also equivalent to that defined using $v^{**}$, as in Eq. (4.5) in Tung (1986). The left-hand term in Eq. (6) is the EP flux pseudo-divergence and the right-hand term in Eq. (6) is the zonal-mean northward flux of PV on the isentropic surface. According to Tung (1986), the PV flux corresponds to the pseudo-divergence of the EP flux along isentropic surfaces and acts as the net eddy forcing term of the mean flow. Note that the pseudo-divergence of the EP flux is used here (instead of the divergence) due to the fact that the density on isentropic surfaces changes with time, which is the main difference from the conventional EP flux (Edmon et al., 1981).

Using the concepts of PV flux and EP flux pseudo-divergence, we rewrite Eq. (3) as

$$\frac{\partial P_a}{\partial t} + \frac{1}{\rho_\theta}\nabla \cdot (\rho_\theta P_a \boldsymbol{V}_c) + \frac{1}{\rho_\theta}\nabla \cdot (\rho_\theta P_c \boldsymbol{V}_a) + \frac{1}{\rho_\theta}\nabla \cdot (\rho_\theta P_a \boldsymbol{V}_a) + P\left(\frac{1}{\rho_\theta}\frac{\partial \rho_\theta}{\partial t}\right) = F_a. \tag{8}$$

The second to fourth terms on the left-hand side are the linear and nonlinear terms of the density-weighted PV flux divergence, while the last term is the local density tendency. Since the density-weighted PV is proportional to vorticity, the second to fourth terms can be interpreted as the dynamical contribution to the PV evolution. On the other hand, since the density tendency is inversely proportional to the temperature tendency, the fifth term can be interpreted as the thermodynamic component of the PV evolution. Equation (8) is derived by converting the advection terms in Eq. (3) into density-weighted flux divergence using the continuity equation. Taking the inner product between Eq. (8) and $E_1$ and neglecting the longitudinal variation of $E_1$ given its wavenumber-0 structure, i.e., $E_1 \approx E_1(\phi)$, we obtain an approximation of the linear and nonlinear terms in equation Eq. (8)

(a detailed derivation is provided in Appendix B):

$$L_{total} \approx 2\pi a^2 \int_{\phi_1}^{\phi_2} E_1(\phi) \frac{\partial [\rho_\theta P_a^{**} v_c^*] + [\rho_\theta P_c^{**} v_a^*]}{\partial \phi} \cos \phi \, d\phi,$$

$$\tag{9}$$

$$N_{total} \approx 2\pi a^2 \int_{\phi_1}^{\phi_2} E_1(\phi) \frac{\partial [\rho_\theta P_a^{**} v_a^*]}{\partial \phi} \cos \phi \, d\phi,$$

where $v$ is the meridional wind, $a$ is the radius of the Earth, and $\phi$ is the latitude with $\phi_1 = 30°N$ and $\phi_2 = 90°N$. Next, we combine the mode equation budget from Eq. (5) with the zonal-mean PV flux from Eq. (6). Based on the relation shown in Eq. (6) and the fact that $\nabla \cdot \mathcal{F} \propto \frac{\partial [u]}{\partial t}$ from the zonal momentum equation, one can further obtain the following relation

$$-\frac{\partial [P^{**} v^*]}{\partial \phi} \propto -\frac{\partial}{\partial \phi} \left( \frac{\partial [u]}{\partial t} \right) \propto \frac{\partial \zeta_\theta}{\partial t}. \tag{10}$$

Equation (10) shows that the meridional gradient of zonal-mean PV flux $\left( -\frac{\partial [P^{**} v^*]}{\partial \phi} \right)$ is connected to the vorticity tendency $\left( \frac{\partial \zeta_\theta}{\partial t} \right)$, which is the dynamical component of the rate of change of $A_1$.

## 3 Results of mode decomposition: mode equation budget

As shown in Figure 1, the first EOF spatial pattern $E_1$ of the PV daily anomalies in ERA-Interim takes the shape of a
260 wavenumber-0 structure with a negative anomaly at the pole. Thus, a positive [1] (negative) value of $A_1$ (PC time series of $E_1$) indicates a weakening (strengthening) of the polar vortex. For example, Figure 2a shows the corresponding $A_1$ for all winter days of ERA-Interim (8490 days in total) and the red vertical lines indicate the onset day of the SSW events. Before SSW events occur, $A_1$ increases significantly and is strongly positive on the onset day, indicating a weakening and breakdown of the polar vortex. Similar EOF spatial patterns are found for the Isca model data (Figure C1 in Appendix C), together with a
265 similar increase in $A_1$ when approaching the SSW central day (Figure 2b). The Isca model data consists of a total of 130 years of simulation, corresponding to 27300 winter days, of which Figure 2b only shows the winter days from the first 40 years as an example. By understanding what contributes to the changes in $A_1$, we extract information that helps explain the breakdown of the polar vortex during SSW development. We compute the mode equation budget of $A_1$ using daily data concatenated for the period -50 to -1 days prior to the onset day for all SSW events. There are a total of 25 SSWs in the ERA-Interim reanalysis data,
and 78 SSWs in the 130-year simulation in the Isca model. Next, we show the composite of SSW events for both reanalysis and the Isca model data.

---

[1]Note that the sign of $E_1$ is not important. Given the pattern of $E_1$, the positive value of $A_1$ corresponds to the weakening of the polar vortex and indication of a potential SSW event.

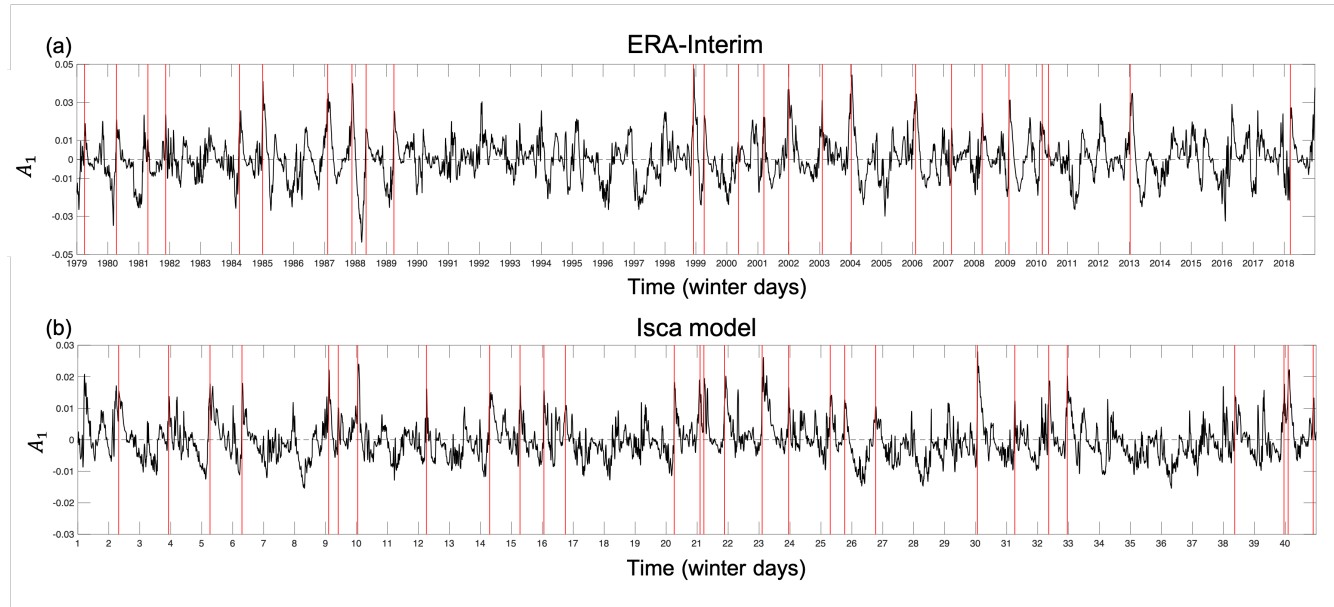

**Figure 2.** The PC time series ($A_1$) of all winter days corresponding to the first EOF mode $E_1$ using (a) ERA-Interim reanalysis data and (b) Isca model output data. In (a) all winter days (from October to April) are shown and in (b) only the winter days for the first 40 years are shown as an example. The PC time series for winter days of the remaining 90 years in the Isca model can be seen in Figure C2 in Appendix C. The red vertical lines indicate the onset dates of SSW events.

### 3.1 Mode equation budget for ERA-Interim

Figure 3 shows the SSW composite of the $A_1$ mode decomposition budget. The first, second, and third rows show the results of the composites of all SSW events, wave-1 SSW events, and wave-2 SSW events, respectively. We apply a 5-days running mean to all lines in Figures 3-6 to remove the high-frequency fluctuations. The bold lines in Figure 3-6 indicate the values that are outside the 2.5th to 97.5th percentile range of normal winter days values, which is computed via a bootstrapping procedure described as follows. In each bootstrap sample, we randomly select with replacement 25 sets of 50 consecutive non-SSW winter days (excluding the 50 days before each SSW) across the different years. We then calculate the mean of these 25 sets of non-SSW days to represent the "composite" of normal winter days. We repeat the bootstrap resampling procedure $B = 1000$ times and compare the SSW composite against the 2.5th and 97.5th percentile of the $B$ bootstrap samples. The same procedure is applied also for the two types of SSWs and for the Isca model data, using the number of SSWs in each dataset as the number of sets of consecutive non-SSW winter days in the bootstrapping. If we only select 50 consecutive non-SSW winter days in December, January, February, and March (months when SSWs occur) to reflect the temporal distribution of SSWs, we obtain qualitatively similar results as when using non-SSW days for all winter months. Here we present the results with the bootstrapping using non-SSW days for all winter months. The reconstructed $A_1$ tendency (black line in Figure 3) is computed

from Eq. (5). The left panels show each term in Eq. (5), while the right panels show the combined effect of all linear (red) and all nonlinear (green) terms without separating the contributions from low and high EOF modes. Figure 3a indicates that around 25 days before the central day of SSWs $\frac{dA_1}{dt}$ starts to increase, and the increase shows a steeper slope around 10 days before the SSW event, leading to a large positive $A_1$ on the central day as shown in Figure 2a. Along with the increase of $\frac{dA_1}{dt}$, $L_{low}$ (red) also increases and is well correlated with the $A_1$ tendency ($r = 0.8$). In fact, $L_{low}$ starts to increase from 35 days before the events but its effect is offset by other terms and it becomes the only contributor to the increase of $\frac{dA_1}{dt}$ at 25- to 15-day leads. The nonlinear term $N_{low-low}$ (green) shows a rapid increase around two weeks before the SSW event and, together with the linear term $L_{low}$, significantly contributes to the changes in $A_1$. The high-frequency components are overall weaker than the low-frequency terms, especially the $N_{high-high}$ (cyan), but the $N_{low-high}$ (orange) has large variations and tends to offset the effect of $L_{low-low}$ at -25 to -15 days before the vortex weakening.

The contributions from each term are different between the composites of wave-1 vs. wave-2 events (Figures 3c and 3e), respectively. The amplitude of $L_{high}$ (blue) is large at around one week before the event, and the nonlinear terms are overall small for wave-1 events (Figure 3c) when compared to wave-2 events (Figure 3e). The amplitude of $N_{low-low}$ is the largest starting one week before the onset for the wave-2 events. To better illustrate the different contributions of linear and nonlinear advection terms in Eq. (5) to the increase of $\frac{dA_1}{dt}$ in the two types of SSW events, we combine all the linear terms ($L_{total}$) and all the nonlinear terms ($N_{total}$) and show the results in the right panels of Figure 3. The linear advection term has the most important contribution from around 25 to 15 days before both SSW event types, and the nonlinear advection term becomes more dominant from day -15 to the onset of the wave-2 SSW events (Figure 3f). On the other hand, the linear advection term plays a central role from day -25 to day 0 for the wave-1 SSWs (Figure 3d). The distinct contributions from the linear and nonlinear advection terms for wave-1 vs. wave-2 events indicate that the processes leading to the vortex breakdown of the two types of SSW events are dynamically different. The simultaneous contributions from linear and nonlinear terms in the all-SSWs composite (Fig. 3a,b) can be viewed as being due to the average over wave-1 and wave-2 SSW events within the composite (Figure 3b). For both types of events, the process captured by the increase of the linear advection term initiates the weakening of the polar vortex around one month before the event and plays a central role until day -10. Around 10 days before the event, the linear (nonlinear) advection term has the dominant contribution for the breakdown of the vortex for the wave-1 (wave-2) events, while the nonlinear (linear) terms are less important or even counteracting the increase of $\frac{dA_1}{dt}$. Therefore, the relative importance of the linear and nonlinear terms emerges as a good indicator of the type of SSW events with a lead time of around 10 days prior to the events.

The relative importance of the linear and nonlinear advection terms for the two types of SSW events is similar to that of the stratospheric wave amplitude of the wavenumber 1 and 2 components for wave-1 and wave-2 SSW events as shown in Bancalá et al. (2012). From their composite analysis of the wave-2 SSW events, the wave-2 component of the geopotential height anomaly in the stratosphere is significantly positive during day -10 to day 0, and the anomalous increase of wave-1 component was found in the period from day -30 to day -10. In Figure 3 we find that the linear and nonlinear advection terms behave similar to the wave-1 and wave-2 components of the upward propagating wave activity. Consistent with the two periods in

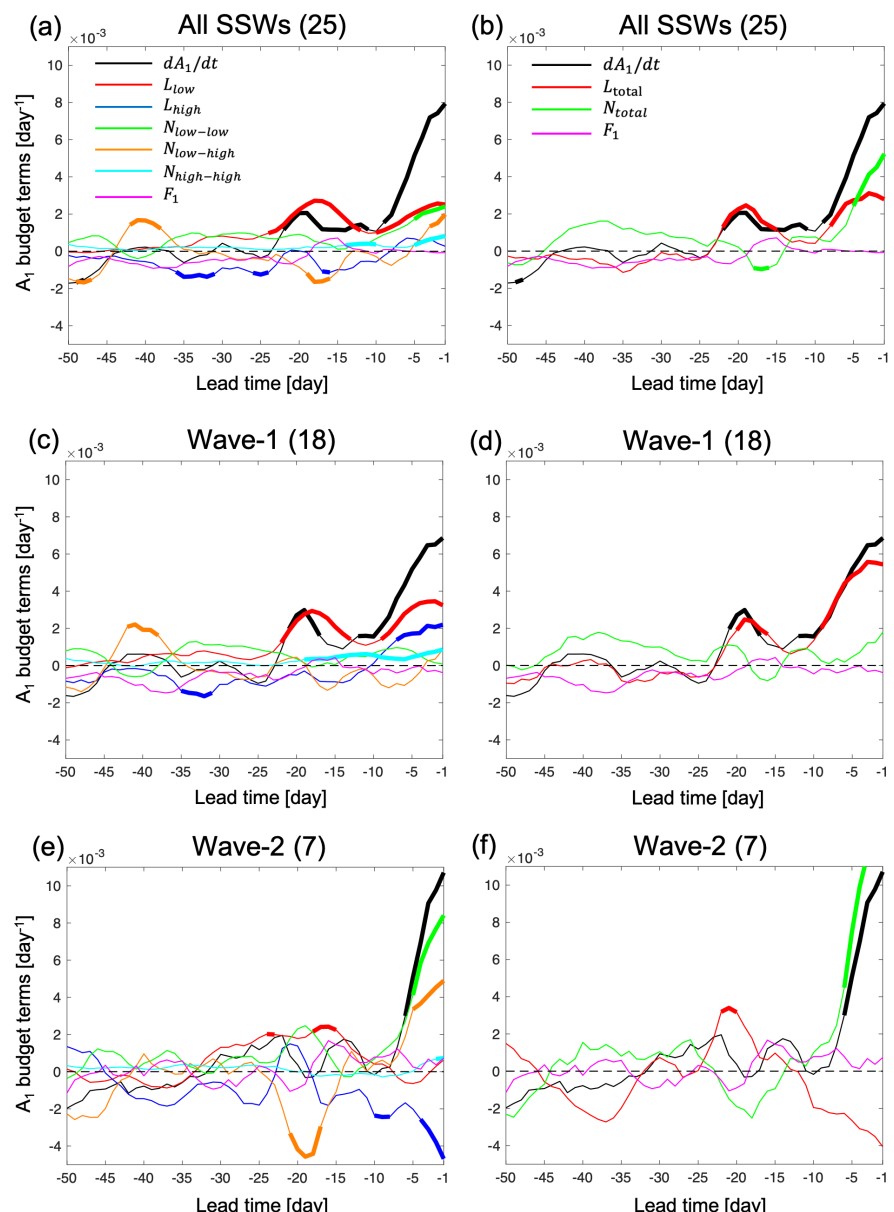

**Figure 3.** The SSWs composite of the $A_1$ budget as a function of lead time from 50 to 1 days before the onset of events in ERA-Interim. (a, b) Composite of all 25 SSW events, (c, d) composite of the 18 wave-1 events, (e, f) composite of the 7 wave-2 events. (a, c, e) show each term of the mode equation budget (Eq. (5)) separately, and (b, d, f) show the total linear term and the total nonlinear term for the different subsets of SSWs. The number in the bracket in each panel title indicates the number of SSW events. A 5-day running mean is applied to all lines. Bold lines indicate values that are outside the 2.5th to 97.5th percentile range of normal winter days values from bootstrapping as described in the text. The representation of each line color in (c,e) and (d,f) is the same as the legend in (a) and (b), respectively.

Bancalá et al. (2012), the evolution of linear and nonlinear terms can be separated into two different time periods, with one from day -25 to day -15 and the other from day -10 to day 0. Since the displacement (split) SSW events are mainly attributed to the enhanced upward propagation of wave-1 (wave-2) (Nakagawa and Yamazaki, 2006; Bancalá et al., 2012), we also look into the contributions of linear and nonlinear terms to $\frac{dA_1}{dt}$ for displacement and split SSW events (shown in Figure D1 in Appendix D), with very similar results. Comparing Figures D1a,b with Figures 3c,d, the behaviors of each term, the total linear term, and the total nonlinear advection term, are very similar as most of the displacement events are wave-1 events (only the event in March 2000 is a wave-2 event). Among the 12 split SSWs, 6 events are dominated by wave-1 wave flux, and most of them do not have a clear split-type behavior as those dominated by wave-2 wave flux. On the other hand, comparing Figures D1c,d to Figures 3e,f, the differences between linear and nonlinear terms are more obvious for wave-2 SSWs as only half of the split events are included in wave-2 events. The behavior of the linear and nonlinear PV advection terms for wave-1 and wave-2 SSWs as shown in Figure 3d and 3f (for displacement and split SSWs, see Figure D1) is comparable to the results in Smith and Kushner (2012), who showed that displacement (split) SSW events are preceded by pronounced linear (nonlinear) vertical wave activity. Our results suggest that the linear and nonlinear contributions are more strongly related to the dominant wavenumber wave forcing than to the vortex geometry.

In order to examine the significance in the differences and the robustness of the relative importance of the linear and nonlinear advection terms for the two types of SSW events, we perform bootstrapping on individual wave-1 and wave-2 events with replacement, respectively. We repeat the resampling $B = 1000$ times and compute the means and the standard deviation for the sum of linear and nonlinear terms. The results of the bootstrapping are shown in Figure 4. There is almost no overlap between the $\pm 1$ standard deviation of wave-1 (red) and wave-2 (black) events in neither the total linear advection (Figure 4a) nor the total nonlinear advection (Figure 4b) term at one week before the onset of the events. In particular, the separation of the wave-1 and wave-2 events in the total linear advection term is as early as 10 days before the onset of the events. Figure 4 demonstrates the significance and robustness of the differences in the contribution of the linear and nonlinear advection terms to $\frac{dA_1}{dt}$ in the wave-1 and wave-2 SSW events, respectively, at least up to one week before the events. Another point to highlight is that significant anomalies of the linear terms are observed around 20 days before both types of SSW events, which is beyond the current predictability limit of SSWs of one-two weeks.

## 3.2 Mode equation budget for the Isca model

Given the limited number of SSW events in the reanalysis data and to further examine the characteristics and robustness of the linear and nonlinear terms contributions to the vortex breakdown, we now apply the same analysis as for ERA-Interim to the output of the Isca model experiment. We use the methodology from Section 2 to extract the EOF modes (spatial patterns) and apply the mode decomposition analysis to the data concatenating 50 to 1 days prior to the 78 SSWs present in the model data. The EOF spatial patterns derived from the model output are similar to those derived from ERA-Interim, especially the first 10 EOF modes (Figure C1), indicating that the model is able to capture the PV features as in the reanalysis.

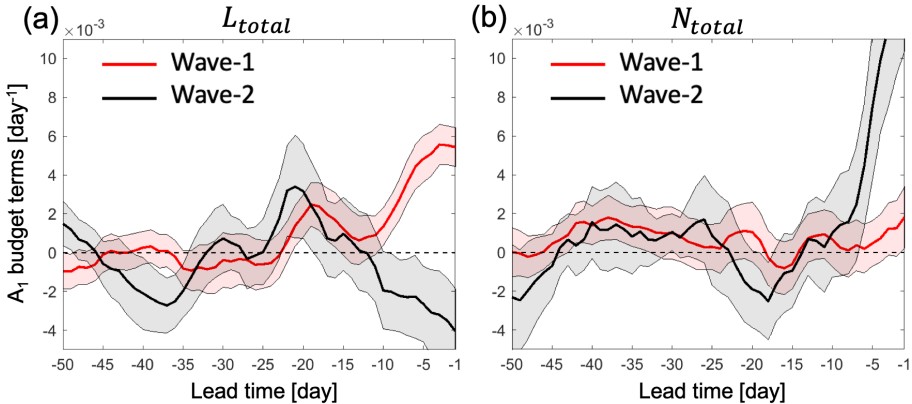

**Figure 4.** The comparison of the $A_1$ budget of the bootstrapping between wave-1 and wave-2 SSW events as a function of lead time from 50 to 1 days before the onset of events in ERA-Interim. (a) Sum of linear advection terms, and (b) sum of nonlinear advection terms for wave-1 events (red) and wave-2 events (black). A 5-day running mean is applied to all lines. Bold lines and the shading indicate the mean and $\pm1$ standard deviation of a bootstrapping using $B =1000$ samples.

Figure 5 shows the results of the $A_1$ budget for the SSW composites. Similar to the results in ERA-Interim (Figure 3), the linear term $L_{low}$ starts to increase at around day -25, but the increase in $\frac{dA_1}{dt}$ starts at around day -10 (Figure 5a), which is later than that in ERA-Interim (around day -25). In the period of day -10 to day 0 of SSWs, $N_{low-high}$ increases rapidly.

The increasing $N_{low-high}$ and $L_{low}$ lead to a rapid increase of $\frac{dA_1}{dt}$. We note that in the 2 days before the onset date, the magnitude of the wave-1 and wave-2 wave fluxes is similar (with differences $< 5Kms^{-1}$) in some SSW events simulated in the Isca model. In order to make a clearer separation between wave-1 and wave-2 SSWs, we exclude events with the difference in the magnitude of wave-1 and wave-2 heat flux at 100 hPa and 60°N smaller than 5 $Kms^{-1}$ as these events cannot be clearly categorized as either wave-1 or wave-2 events. Therefore, in the model we have 36 wave-1, 27 wave-2, and 15 unclassified

events. Figure 5c shows that $L_{low}$ for SSWs increases and starts to differentiate from normal winter days starting at a lead of 20 days for wave-1 SSWs. Different from the $A_1$ budget of wave-1 SSWs in ERA-Interim (Figure 3c), $N_{low-high}$ starts to increase from day -7 and becomes an important contributor to $\frac{dA_1}{dt}$. For wave-2 SSWs, Figure 5e shows that $N_{low-high}$ is the main contributor to the increase of $\frac{dA_1}{dt}$ from day -10, and $N_{low-low}$ as well as $L_{high}$ are the second largest contributors to $\frac{dA_1}{dt}$ from day -5, which is different from the evolution of $L_{high}$ for wave-2 SSWs in ERA-Interim (Figure 3e). In both types

of events, $L_{low}$ starts to increase at around day -20, which helps to weaken the polar vortex in the preconditioning stage and is similar to the evolution of $L_{low}$ (with a smaller amplitude) in the same period in ERA-Interim. The effects of $L_{total}$ and $N_{total}$ are shown in the right panels in Figure 5. The evolution of $L_{total}$ and $N_{total}$, and thus of $\frac{dA_1}{dt}$, for the Isca model (Figure 5b) is similar to the evolution of these terms for the ERA-Interim data (Figure 3b), indicating that the Isca model successfully reproduces the vortex breakdown in the 10 days preceding the SSWs. The increase of $\frac{dA_1}{dt}$ can therefore be used to predict

the occurrence of SSWs with one-two weeks lead time. Even though the distinct increase of $\frac{dA_1}{dt}$ only shows up at around

day -10, $L_{total}$ actually increases as early as day -29. However, this amplification in the linear term is offset by the nonlinear and forcing terms, which leads to a near-zero $\frac{dA_1}{dt}$. Figures 5d and 5f show the evolution of $L_{total}$ and $N_{total}$ of the wave-1 and wave-2 SSW composites, respectively. Different from wave-1 events in ERA-Interim, the linear and nonlinear terms are equally important. However, when comparing wave-1 with wave-2 composites, one can still see the difference in the relative
importance of the linear and nonlinear terms for the two types of SSWs. The nonlinear term is stronger in wave-2 SSWs than in wave-1 SSWs and is more than twice as large as the linear term one week before the central day (Figure 5f). Thus, our finding from the ERA-Interim that the linear (nonlinear) term is important for wave-1 (wave-2) events is also true for the Isca model data. The main differences compared with the reanalysis data are that $\frac{dA_1}{dt}$ exhibits a substantial increase only from day -10, and the variations of all terms in Eq. (5) are overall small before day -10, which could potentially limit the predictability of
SSWs in the Isca model. Different reasons might be able to explain the differences in results between the Isca model and the reanalysis, e.g., different model complexities (e.g., lack of parameterizations for gravity waves breaking and interactive ozone chemistry in the Iscal model) or the coarse model horizontal resolution (T42), both of which might lead to an underestimation of some of the high-frequency variability in comparison with the reanalysis. Another important difference is that in the Isca model the nonlinear term displays larger values for both types of SSWs compared to the reanalysis. This latter behavior might
be related to the stronger SSW-sensitivity to wave-2 forcing in the idealized models (e.g., Iscal model) where most of the SSW are likely triggered by wave-2 activity (Gerber and Polvani, 2009), in contrast with wave-1 which seems to be more dominant in more complex general circulation models or reanalysis datasets (see for example Figure 3 in Ayarzagüena et al., 2018). Note that even when classifying an SSW event as a wave-1 event in the model, its wave-2 component, although weaker than the wave-1 component, might still play an important role in the overall evolution of the event.

**4   Physical interpretation of the mode decomposition**

In our analyses above, we found that the persistent positive values of $\frac{dA_1}{dt}$ and its contributors that emerge during the vortex breakdown, such as during SSWs, are significantly different from the values observed during normal winter days. Additionally, the signals identified as representative of wave-1 and wave-2 events are also different. We also observed that the signals that are characteristic of SSWs emerge as early as 20-25 days before the onset of SSWs. Given that these results hint that SSWs
are potentially predictable at longer lead times, i.e., beyond the current predictability limit of one-two weeks, in this section we provide a physical interpretation of these signals that we identified through the mode decomposition analysis.

As demonstrated in Section 2.4, the total linear and nonlinear advection terms in Eq. (5) are closely linked to the PV flux divergence, which offers a more intuitive interpretation in an Eulerian framework. In this section, we use ERA-Interim data to illustrate the physical interpretation of the increase of the linear and nonlinear advection terms in Eq. (5). The motivation to
400 introduce the PV flux into the PV equation is that the zonal-mean PV flux is connected to the pseudo-divergence of the EP flux along isentropic surfaces, and thus acts as the net eddy forcing term of the mean flow, thus allowing for connections with the

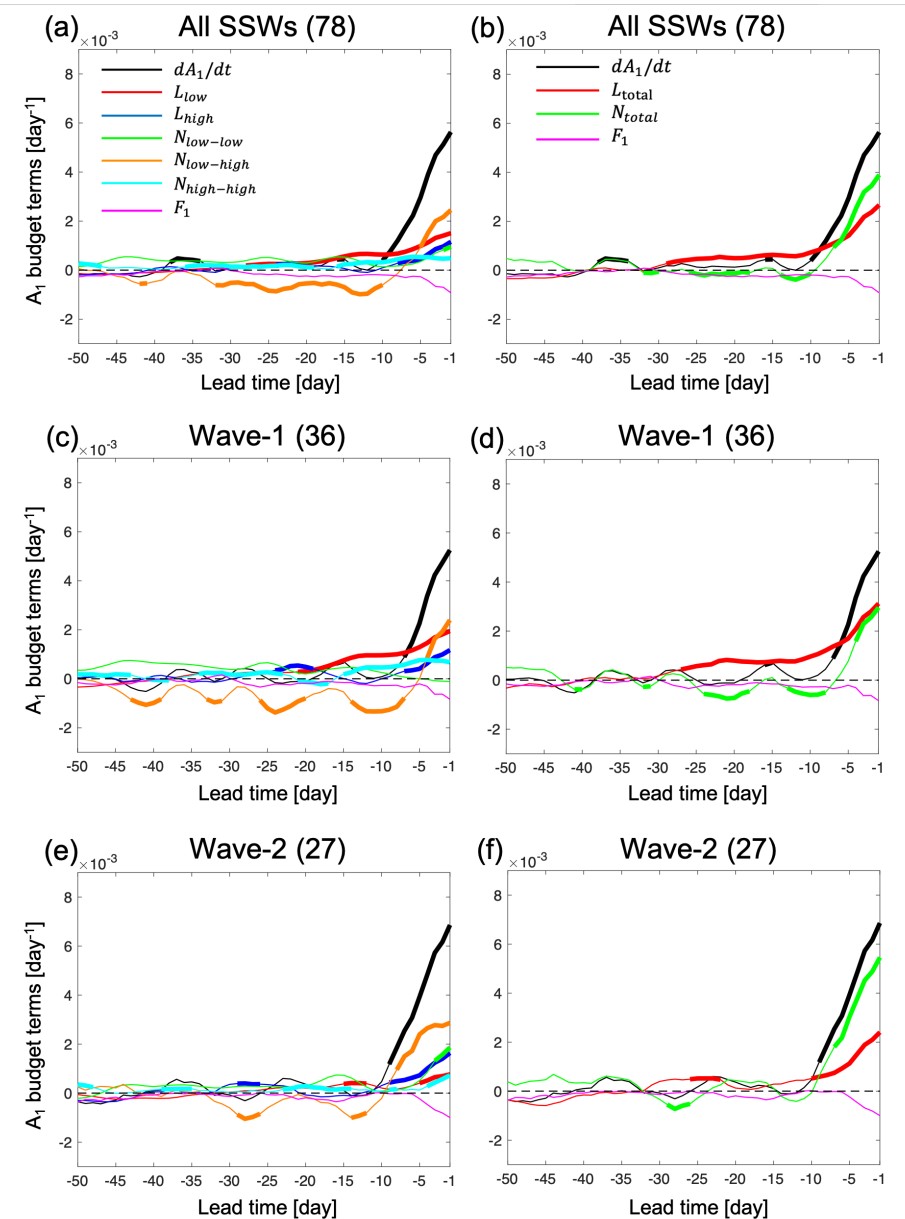

**Figure 5.** The composite of $A_1$ budget as a function of lead time from 50 to 1 days before the onset of the events in the Isca model output. (a, b) Composite of the total 78 SSW events, (c, d) composite of the 37 wave-1 events, and (e, f) composite of the 26 wave-2 events. (a, c, e) show each term of Eq. (5) separately, and (b, d, f) show the total linear term and the total nonlinear term for the different subsets of SSWs. All lines are smoothed by a 5-day running mean. Bold lines indicate the values that are outside the 2.5th to 97.5th percentile range of normal winter days values from bootstrapping as described in the text. The representation of each line color in (c,e) and (d,f) is the same as the legend in (a) and (b), respectively.

theory of wave-mean flow interaction (i.e., Eq. (6) and Eq. (10)). According to McIntyre and Palmer (1983), the wave activity of planetary waves is converted to the angular momentum of the mean flow, which violates the non-acceleration condition (Charney and Drazin, 1961), leading to the reversal of the mean flow. To understand the importance of the zonal-mean PV flux during the development of SSWs, we decompose the zonal-mean PV flux into different components as in Ayarzagüena et al. (2011),

$$[\rho_\theta][\rho_\theta P^{**} v^*]\cos\phi = [\rho_\theta]\left([\rho_\theta P_c^{**} v_c^*] + [\rho_\theta P_c^{**} v_a^*] + [\rho_\theta P_a^{**} v_c^*] + [\rho_\theta P_a^{**} v_a^*]\right)\cos\phi, \tag{11}$$

where the subscript $c$ represents daily climatology, and $a$ represents daily anomalies. On the right-hand side of Eq. (11), the first term corresponds to the climatological planetary waves, the second and third terms correspond to the interaction between the climatological planetary waves and the daily anomalies, and the fourth (last) term corresponds to the interaction between daily anomalies. Similar to Eq. (3), the second and third right-hand terms can be viewed as linear components, and the fourth term as the nonlinear component. Figure 6 shows the composite of zonal-mean poleward PV flux averaged north of 45°N and its decomposition in Eq. (11) as a function of lead time ahead of SSW events. The first term on the right-hand side of Eq. (11) can be seen as a constant since its variation with time is very small as shown in Figure 6. When approaching the onset day of SSWs, the zonal-mean PV flux (black) becomes increasingly negative, indicating a weakening of the polar vortex. This further decrease of the negative zonal-mean PV flux is mainly due to the linear interaction between the climatological planetary waves and the daily anomalies (red) and the nonlinear interaction between anomalies (green), which correspond to the total linear and nonlinear PV advection terms (right columns in Figure 3), respectively. Even though the climatological planetary waves (blue) also have negative contribution to the total PV flux, the variations are very small with time. Similar to the distinct contributions of the linear and nonlinear PV advection terms in the wave-1 and wave-2 SSW composites, the negative total zonal-mean PV flux is mainly due to its linear component in wave-1 SSWs (red in Figure 6b) and its nonlinear component in wave-2 SSWs (green in Figure 6c). The different behavior of the linear and nonlinear PV flux during different types of SSW events is consistent with the behavior of PV advection in Figure 3 and is well aligned with the behavior of the vertical wave flux as shown in Figures 7 and 8 of Smith and Kushner (2012). Different from Smith and Kushner (2012), the nonlinear (linear) component of the PV flux even becomes positive just before the onset of wave-1 (wave-2) events (the positive linear PV flux in wave-2 is statistically different from normal winter days), counteracting the weakening of the polar vortex. Similar behavior can also be found in the linear advection term for wave-2 SSWs composite (Figure 3f). Even though the amplitude of the negative linear PV flux at around day -15 in wave-2 SSWs composite is small, it helps to weaken the polar vortex and to offset the effect induced by the positive nonlinear PV flux. Note that it is the meridional gradient of the poleward zonal-mean PV flux that is used to approximate the linear and nonlinear PV advection terms as demonstrated in Eq. (9). The poleward zonal-mean PV flux is proportional to the magnitude of its meridional gradient and one can thus use it to approximate the linear and nonlinear advection terms and to provide a physical interpretation of the signals found in Section 3.

From Figure 6, the linear and nonlinear zonal-mean PV flux emerge as potential indicators for the type of SSW events. Given the abrupt change of the PV flux in the 10 days preceding the onset of the events, we next examine how the spatial patterns of the poleward PV flux lead to their distinct zonal-mean contributions in the two types of SSWs. Figure 7 shows the poleward

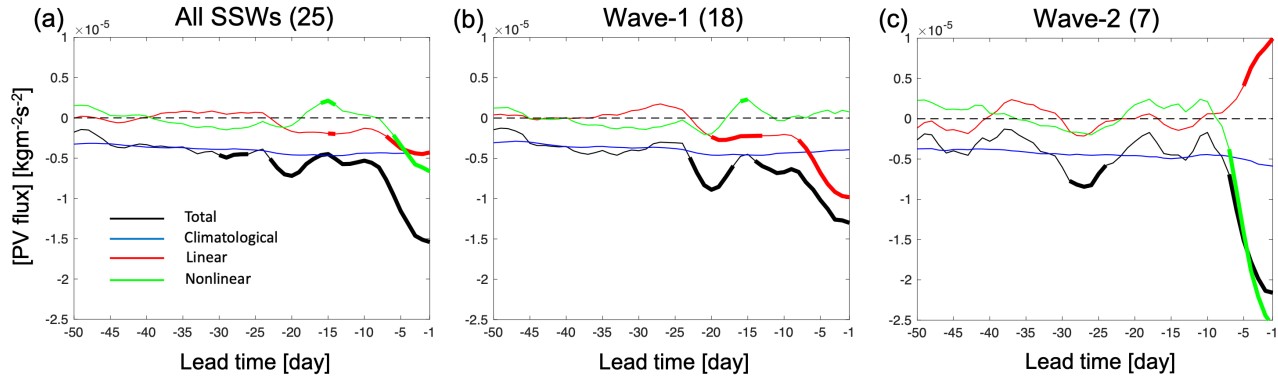

**Figure 6.** Composites of zonal-mean poleward PV flux averaged north of 45°N and its decomposition in Eq. (11) as a function of lead time before SSWs: (a) all SSW events, (b) wave-1 events, and (c) wave-2 events in ERA-Interim. A 5-day running mean is applied to all lines. Bold lines indicate the values that are outside the 2.5th to 97.5th percentile range of normal winter days values from bootstrapping as described in the text. Subfigures (a-c) all share the same color legend as (a).

linear and nonlinear PV flux horizontal patterns of the wave-1 and wave-2 composites. As shown in the previous analyses, the linear signals start to amplify at around 20-25 days preceding the onset of both types of SSWs, and later the nonlinear signals become the most important contributors to the breakdown of the vortex from day -10 onwards for wave-2 events, while the linear signals keep amplifying for wave-1 events. Based on the different behaviors of linear and nonlinear terms, the deceleration of the polar vortex can be separated into two different periods: the first period from around day -25 to day -15 and the second period from day -10 to day -1. Thus, two different lead times (day -15 and day -5) are displayed in Figure 7 as example to illustrate the spatial pattern of the PV flux in the two different periods before the onset of the events. The spatial patterns for the linear PV flux of wave-1 events are quasi-stationary from around day -20 (Figures 7a and 7b). A similar wave-2 pattern is also shown in linear PV flux for wave-2 events in the period of 28 to 12 days preceding the onset date (Figure 7c). This pattern disappears from day -11 (not shown), and the wave pattern shown in Figure 7d develops continuously until the central day of the SSW event. At the same time the positive values of PV flux increase, leading to a positive zonal-mean PV flux when close to day 0. Different from the linear PV flux, the nonlinear PV flux shows a clearly higher wavenumber pattern in the second period of the development of wave-2 SSWs shown in Figure 7h, which has a strongly negative PV flux over western North America and a strongly positive PV flux over eastern North America. The downwind growth of the PV flux reaches its minimum (negative anomalies) over the North Atlantic and shows positive anomalies over the Northern Europe. Then the PV flux gradually weakens downstream over North Asia and the North Pacific. This organized wave pattern in the nonlinear PV flux does not emerge until day -11 and remains largely stationary until the onset of the wave-2 events. In the early stage of the warming (day -25 to day -15), the nonlinear PV flux has a very low magnitude as shown in Figure 7g. Since previous studies suggested that split SSWs have a predominantly barotropic structure (Manney et al., 1994; Matthewman et al., 2009; Albers and Birner, 2014), we investigate the vertical structure of the nonlinear PV flux in the wave-2 SSWs composite. Figure

8b shows the longitude-height cross section of the nonlinear PV flux of the wave-2 SSWs composite at day -5 as an example. As can be seen in Figure 8b, the wave pattern shown in Figure 7h extends throughout the stratosphere and displays barotropic characteristics for wave-2 events. On the other hand, the longitude-height cross section of the linear PV flux at day -5 of the wave-1 SSWs in Figure 8a displays a more baroclinic structure in the Eurasia and Pacific regions. The spatial pattern of the nonlinear PV flux in the composite of wave-1 events exhibits substantial transient fluctuations without a clear wave pattern before day -9 (Figure 7e), and shows a more stable and organized spatial pattern in the period from day -9 to day 0 (Figure 7f). However, the magnitude of the nonlinear PV flux in wave-1 events is smaller than its linear flux counterpart and also smaller than the nonlinear PV flux in the wave-2 events composite.

Even though the spatial patterns of the linear PV flux in the two types of SSWs show a wave-2 pattern in the period of 20 to 10 days preceding the central day of SSWs, the locations of the maximum and minimum PV flux shift around $30°$ in longitude in the Pacific and North America regions (Figures 7a and 7c). In the first period from day -10 to day 0, the wave pattern shown in Figure 7h sets in and leads to the final split of the vortex for wave-2 SSWs. One relevant question is what processes lead to the differences observed in the evolution of the vortex breakdown where the linear PV flux remains important for wave-1 events, while the nonlinear PV flux amplifies for wave-2 events. Some previous studies suggested that the pre-SSW evolution of the polar vortex is distinct between split and displacement events (Charlton and Polvani, 2007; Bancalá et al., 2012), and this preconditioning could trigger the nonlinear resonance of planetary waves in the lower stratosphere, leading to the split of the polar vortex (Albers and Birner, 2014; Boljka and Birner, 2020). Here we examine the anomalies of PV and meridional wind after removing the daily climatology and zonal mean ($P_a^{**}$ and $v_a^*$, respectively, see Eq. (11)) for the wave-1 and wave-2 events to understand their distinct evolutions after day -10. Figure 9a-b show the spatial pattern of $P_a^{**}$ (shading) and $v_a^*$ (green contour) at day -15 before wave-1 and wave-2 events, respectively. Both $P_a^{**}$ and $v_a^*$ present wave-1 patterns, but the positive and negative anomalies are located in different regions. The whole pattern of $P_a^{**}$ in wave-2 SSWs is around $60°$ further east compared to wave-1 SSWs. The negative $v_a^*$ is mainly located over eastern North America and the northern North Atlantic, which is important for the negative PV flux in the same region (Figure 7a) for wave-1 SSWs, while for wave-2 SSWs the negative $v_a^*$ covers all of North America. These differences in the location of $P_a^{**}$ (shading) and $v_a^*$ (green contour) between the two types of SSW events are amplified from day -10 (Figure 9c-d). The magnitudes of $P_a^{**}$ and $v_a^*$ in the first period from day -10 to day 0 are larger than in the second period from day -25 to day -15. The negative $P_a^{**}$ is located more over North America, while the positive $P_a^{**}$ is located over the North Atlantic and Europe for wave-1 SSWs (Figure 9c). For wave-2 SSWs, the pattern of $P_a^{**}$ is the opposite (Figure 9d). The positive $v_a^*$ extends to the full North Pacific (Figure 9c) and both $P_a^{**}$ and $v_a^*$ maintain wave-1 structure for wave-1 SSWs. In contrast, $v_a^*$ (Figures 9d) develops a wave-2 structure from day -10 onwards for wave-2 events. The weak positive $P_a^{**}$ over Asia (Figure 9d) further develops from day -5, resulting in a wave-2 structure over mid-latitude at day 0 (not shown). The main features of nonlinear PV flux in Figure 7 can be roughly inferred by $P_a^{**}$ and $v_a^*$ in Figure 9. We note that the composite of the nonlinear PV flux term is not equal to the direct product of the composites of $P_a^{**}$ and $v_a^*$, and thus some features in Figure 7 cannot be directly inferred from Figure 9. We also find that the nonlinear PV flux and the $P_a^{**}$ and $v_a^*$ form a positive feedback from around day -10 to the onset of the wave-2 events. As the amplitude of

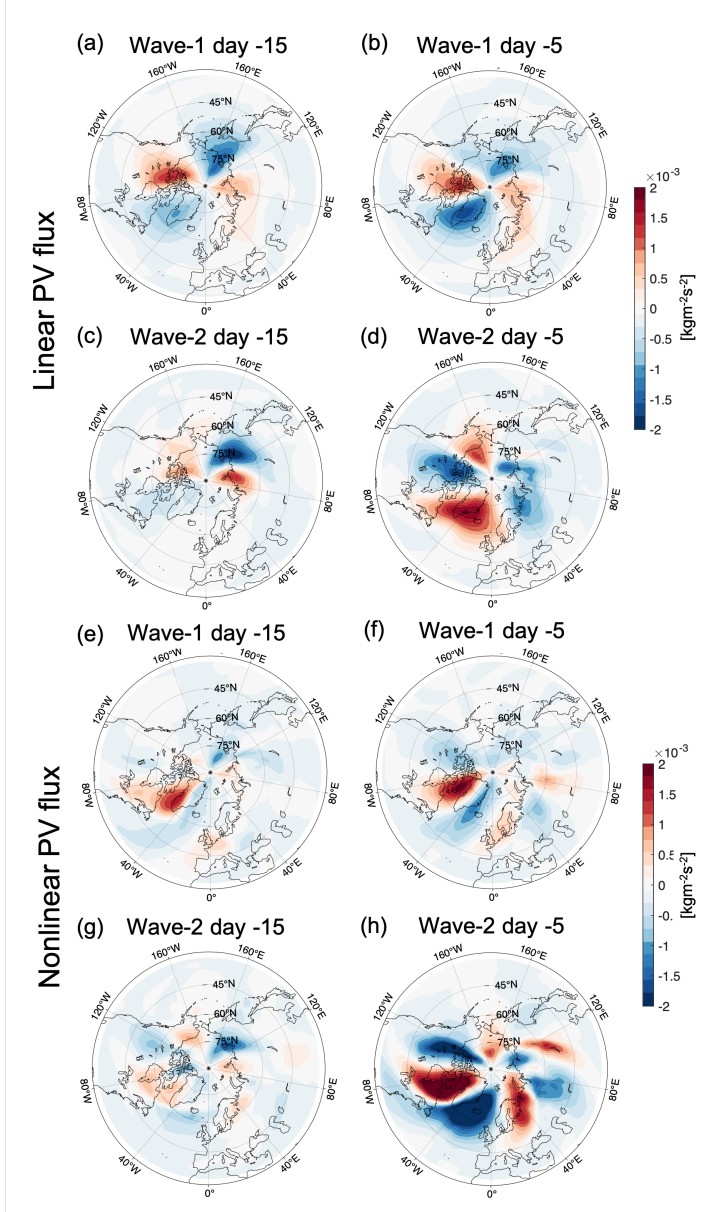

**Figure 7.** The spatial pattern of the linear PV flux for (a, b) the composite of wave-1 SSWs, and (c, d) the composite of wave-2 SSWs. The spatial pattern of the nonlinear PV flux for (e, f) the composite of wave-1 SSWs, and (g, h) the composite of wave-2 SSWs. (a, c, e, g) show the spatial pattern at day -15, and (b, d, f, h) show the spatial pattern at day -5 prior to the events.

$P_a^{**}$ and $v_a^*$ becomes larger, the nonlinear PV flux is also amplified, particularly in the region of the negative nonlinear PV flux over western North America and the North Atlantic as shown in Figure 7h. The strong negative nonlinear PV flux contributes to more negative net zonal-mean PV flux values, suggesting a zonal-mean EP flux convergence. This EP flux convergence thus further decelerates the polar vortex. According to the non-acceleration theorem (Charney and Drazin, 1961), the deceleration of the polar vortex is accompanied by stronger wave activity, which is represented by the increasing amplitude of $P_a^{**}$ and $v_a^*$.

We also note that the spatial pattern of $P_a^{**}$ in Figure 9f finally leads to the split of the polar vortex in wave-2 events, with the positive values corresponding to one of the daughter vortices located around $60^\circ$W.

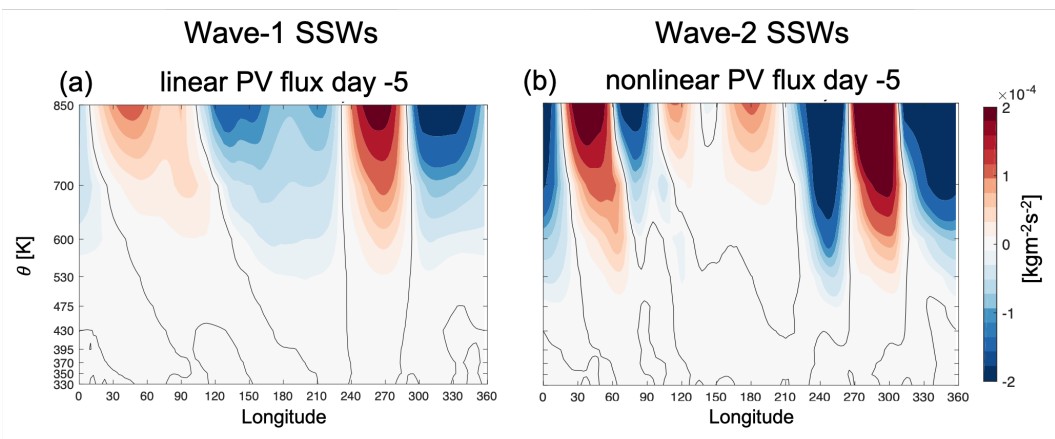

**Figure 8.** The longitude-$\theta$ cross section of PV flux averaged north of $45^\circ$N at day -5. (a) Linear PV flux for wave-1 SSWs and (b) nonlinear PV flux for wave-2 SSWs. Black line indicates the zero value of PV flux.

## 5   Conclusions

In this paper we employ a mode decomposition analysis to investigate the preconditioning of sudden stratospheric warming events. We study the (linear and nonlinear) terms in the potential vorticity equation by means of a budget analysis in order

to identify the components in the first PC time series $A_1$ that allow us to distinguish the behavior of the polar vortex during SSW events from normal winter days. Moreover, we identify characteristics of SSWs that help to identify the type of event (wave-1 vs. wave-2) during the dynamical development of SSWs. The mode decomposition analysis allows us to obtain a mode equation budget that describes the temporal evolution of the stratospheric dynamical processes that lead to the breakdown of the polar vortex. A better understanding of the vortex weakening process may help to improve the predictability of SSW events.

The rate of change of the first PC time series $\left(\frac{dA_1}{dt}\right)$ represents the evolution of the strength of the polar vortex, and we find a significant increase in $\frac{dA_1}{dt}$ at around 25 days before the onset of SSWs. This change in $\frac{dA_1}{dt}$ marks the start of the vortex weakening process indicating an acceleration of the polar vortex breakdown, and is different from the evolution of $\frac{dA_1}{dt}$ during normal winter days. The lead time of 25 days that we identified in our analysis is far beyond the current predictability limit of

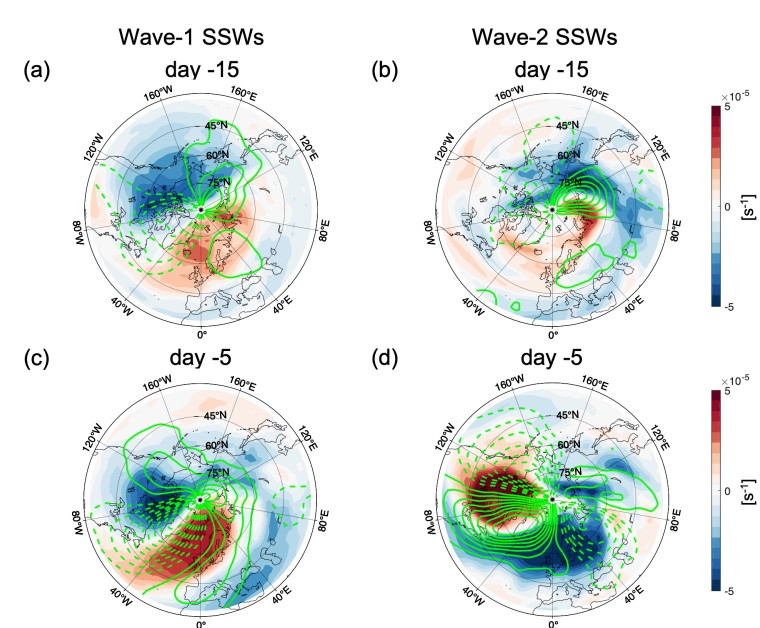

**Figure 9.** The spatial pattern of anomalies of PV and meridional wind after removing the daily climatology and zonal mean values (a,b) on day -15; (c,d) on day -5. (a, c) show the composites of wave-1 SSWs, and (b, d) show the composites of wave-2 SSWs. Shading is for the PV anomalies ($P_a^{**}$) and the green contour is for the meridional wind anomalies ($v_a^*$) with dashes for negative values. The contour interval of $v_a^*$ is 10 m/s.

SSW events (Domeisen et al., 2020b). We note that not only the composite of SSWs, but most of the individual SSW events
show the increase of $\frac{dA_1}{dt}$ at around 20-25 days before the onset. The increase of $\frac{dA_1}{dt}$ is mainly due to the increase of the linear PV advection term, which preconditions the weakening of the polar vortex. While recent work suggests that split SSW events are less predictable than displacement events (Taguchi, 2018; Domeisen et al., 2020b), the preconditioning by the increase in the linear PV advection and PV flux is important for both wave-1 and wave-2 SSW events at around 20 days before the onset of the event, implying a similar intrinsic predictable time scale with 20 days lead for both types of events. From around 10
515 days before the events, the nonlinear PV advection term increases rapidly for wave-2 SSWs, but it remains small for wave-1 SSWs. As the nonlinear PV advection term increases, the linear PV advection drops dramatically prior to wave-2 SSWs. The distinct behavior of the linear and nonlinear advection terms in this 10-day period suggests that the type of SSW event could be inferred at around 10 days to one week prior to the events. Note that the type of event is determined by the larger wavenumber component of eddy heat flux in the period of -2 to 0 days before the events and hence the 10-day lead times
520 need to be interpreted with caution. The above differences are also present in the displacement and split SSW events, but the differences are somewhat smaller than those between wave-1 and wave-2 events, as not all split events are induced by wave-2 planetary waves.

Even though the contributions from linear and nonlinear PV advection terms are different in the two types of SSW events, their overall effects on $\frac{dA_1}{dt}$ within 10 days before the events are the same, causing $\frac{dA_1}{dt}$ to increase abruptly. The breakdown of
525 the polar vortex can be divided into two periods based on the different behavior of the linear and nonlinear terms. During the first period, i.e., from 25 days to two weeks before the onset of SSWs, the linear term weakens the polar vortex for both types of SSWs, and during the second period, i.e., from around 10 days before the onset date until the onset, the vortex evolution for both types of SSWs starts to diverge and the distinct vortex breakdown structures gradually develop. These two different time periods before the onset of the events are consistent with previous studies, especially for wave-2 SSWs (Labitzke, 1981;
Bancalá et al., 2012; Albers and Birner, 2014), which suggested that an amplification of the wave-1 component allows the wave-2 wave flux to grow and propagate more effectively into the already weakening polar vortex region.

In both the ERA-Interim reanalysis and the simplified Isca model experiments, the increase of $\frac{dA_1}{dt}$ is more abrupt for wave-2 SSW events than for wave-1 events and thus results in a larger $\frac{dA_1}{dt}$ in the 10-day period preceding the onset of the events. The abrupt changes in $\frac{dA_1}{dt}$ are mainly due to the exponential increase in the nonlinear PV advection term. By contrast, the
535 linear PV advection for wave-1 SSWs increases more slowly but consistently. The rapid growth of the nonlinear process for wave-2 SSWs could be related to a positive feedback between the nonlinear PV flux and the anomalies of PV and meridional wind when we tried to interpret the underlying dynamics of the increase in the nonlinear PV advection obtained from the mode equation budget. The linear and nonlinear advection terms are closely linked to the PV flux divergence, while the zonal-mean PV flux can be directly related to the zonal mean momentum budget (McIntyre and Palmer, 1983; Tung, 1986; Plumb, 2010).
The zonal-mean poleward PV flux can be further decomposed into the linear and nonlinear components, whose role in the weakening of the polar vortex is similar to the effect of the PV advection terms on the increase of $\frac{dA_1}{dt}$. The wave-2 spatial pattern of the linear PV flux helps to precondition the stratospheric basic state and decelerate the polar vortex in the first period of the SSW development for both types of SSWs. When the vortex weakening process evolves to the second period, the evolution of the PV flux for the two types of SSW events bifurcates as the linear and nonlinear PV flux exclusively amplify in
the wave-1 and wave-2 events, respectively. This bifurcation could be due to the specific evolution of the stratospheric states in the two types of events (Charlton and Polvani, 2007; Albers and Birner, 2014), which can be seen from the horizontal patterns of the PV and meridional wind zonal anomalies. Our results suggest that the high wavenumber pattern that emerged in the second period for wave-2 SSWs is closely connected to the wave-2 wave flux and could be essential to the split of the vortex.

As suggested by Aikawa et al. (2019), mode decomposition analysis allows us to investigate the contribution of each EOF
mode to the breakdown of the polar vortex, and the way the associated spatial patterns play a role in the temporal evolution of the first PC time series. We found that the interactions involving the low modes are the dominant contributors to the weakening of the polar vortex, especially for the increase of the linear advection term in the first period (i.e., day -25 to day -15). Further investigation of the contribution from each EOF mode to the linear and nonlinear advection terms in the mode equation budget suggests that the increase of the linear advection term in the first period is largely influenced by the second and third EOF
modes, which both show a wave-1 structure. The first EOF mode only plays an important role at around one week before the SSW event, suggesting that the process for the vortex weakening is initiated by modes that are not zonally symmetric. In terms

of the nonlinear advection terms, the interactions amongst the first five EOF modes are important when approaching the onset of SSWs.

Even though the increase in $\frac{dA_1}{dt}$ and the contribution from the linear term start at around 25-20 days before the onset of
SSWs in ERA-Interim, one needs to be cautious about the interpretation. The signals shown in the composite of SSWs do not necessarily indicate that these signals can be used to predict each individual SSW event with lead times of 20-25 days. While most SSW events do show a consistent positive $\frac{dA_1}{dt}$ starting 20 days before the SSW onset, around 30% of all SSWs in ERA-Interim do not show this clear increase in $\frac{dA_1}{dt}$ around lead times of 20-25 days. On the other hand, the linear signals found here with lead times of 3-4 weeks may not be exclusive to SSWs. There are cases where large positive values of the first PC time
series do not correspond to an SSW event, but instead to a strong deceleration event. However, $\frac{dA_1}{dt}$ and the contribution from its linear term for SSW events are overall stronger and more persistent than that for the strong deceleration events (not shown). What we found here suggests that the intrinsic predictability of SSWs may be longer than the current two-week practical predictability. However, more work is still needed to investigate whether the practical predictability of SSWs can actually be extended and if yes, how.

Since the signals shown here indicate that most SSWs may be predictable on subseasonal time scales, it is important to understand which processes lead to the variability of the first EOF pattern and help to improve subseasonal forecast skill. A recent study by Albers and Newman (2021) identified two modes that relate to the linear and nonlinear processes for strong downward propagating stratospheric anomalies, with one mode representing purely stratospheric processes and the other mode representing stratosphere-troposphere coupling. However, they point out that it is not clear which processes are
more important for subseasonal predictability. Even though we have not connected the specific physical processes to the linear and nonlinear signals that are important for the two types of SSWs in this study, the spatial patterns of the PV flux could potentially provide some hints for future investigation. For example, the wave-2 spatial patterns of the linear PV flux are relatively stationary for wave-1 SSW events. This is due to the fact that the orientation of the negative and positive anomalies does not change significantly from day -19 onwards and the orientation of the wave-1 structure also remains stationary in both
PV and meridional wind anomalies. These persistent spatial patterns and the linear behaviour in the early stage of development of SSWs may be related to weather phenomena, such as blocking, teleconnections, and low frequency modes in the troposphere. For example, Smith and Kushner (2012) and Cohen and Jones (2011) suggested that displacement events are preceded by sea level pressure anomalies associated with the Siberian high which is consistent with the increase of linear vertical wave flux before the events.

In conclusion, our study finds signals that are representative of SSW events as early as 25 days preceding the events. This lead time is significantly longer than the current predictability limit of SSWs. We furthermore find that mode decomposition analysis can help infer wave-1 and wave-2 events at least one week ahead of the event, which is longer than the lead times identified in previous studies (Karpechko, 2018; Taguchi, 2018; Domeisen et al., 2020b). The time scale of emergence of the distinct evolution between linear and nonlinear terms provides insights into the different dynamical processes responsible for

the two types of SSWs, and thus could be potentially used as a predictor of the type of event in future studies. Given that the noticeable increase in $\frac{dA_1}{dt}$ in the simplified GCM (Isca model) experiment, which directly indicates the weakening of the polar vortex, shows up only around 10 days before the onset of SSWs (i.e., at shorter lead times than for reanalysis), suggests that the observed atmosphere tends to be more predictable than the model, which agrees with theory (Smith et al., 2016; Scaife and Smith, 2018). Applying the mode decomposition analysis to more complex forecasting models, i.e., S2S reforecast models

(Vitart et al., 2017), to examine the predictability of SSWs will provide further insights into the dynamics of the polar vortex weakening and might potentially allow for the prediction of these events beyond the current lead times.

*Code and data availability.* The Isca modeling framework was downloaded from the GitHub repository (https://github.com/ ExeClim/Isca) (Vallis et al., 2018)). ERA-Interim reanalysis (Dee et al., 2011) was obtained from the ECMWF server (https://apps.ecmwf.int/datasets/data/interim-full-daily).

**Appendix: A. Mode decomposition budget equation**

In this appendix, we show the derivation procedure for obtaining the mode decomposition equation budget (Eq. (4)) and the cumulative explained variance and the power spectrum of the first 1000 EOF modes. The spatial patterns associated to the projections ($\boldsymbol{U}_1, \boldsymbol{U}_2, \ldots, \boldsymbol{U}_d$) of the wind vector daily anomalies $\boldsymbol{V}_a$ onto the PC time series are computed by projecting $\boldsymbol{V}_a$ onto the PC time series ($\{A_1, A_2, \ldots, A_d\}$). Note that $\boldsymbol{U}_1, \boldsymbol{U}_2, \ldots, \boldsymbol{U}_d$ are not necessarily orthogonal. The anomaly terms of PV

and the wind vector fields can be written as

$$
\begin{aligned}
P_a &= \sum_{n=1}^{d} E_n A_n, \\
\boldsymbol{V}_a &= \sum_{n=1}^{d} \boldsymbol{U}_n A_n.
\end{aligned}
\tag{A1}
$$

By substituting Eq. (A1) into Eq. (3), we get

$$
\sum_{n=1}^{d} E_n \frac{dA_n}{dt} = -\boldsymbol{V}_c \cdot \sum_{n=1}^{d} \nabla E_n A_n - \left( \sum_{n=1}^{d} \boldsymbol{U}_n A_n \right) \cdot \nabla P_c - \sum_{n=1}^{d} \boldsymbol{U}_n A_n \cdot \sum_{n=1}^{d} \nabla E_n A_n + F_a.
\tag{A2}
$$

Taking the inner product between Eq. (A2) and a given EOF mode $E_k$, we obtain

$$
\sum_{n=1}^{d} \langle E_k, E_n \rangle \frac{dA_n}{dt} = -\sum_{n=1}^{d} \langle E_k, \boldsymbol{V}_c \cdot \nabla E_n \rangle A_n - \sum_{n=1}^{d} \langle E_k, \boldsymbol{U}_n \cdot \nabla P_c \rangle A_n - \sum_{m=1}^{d} \sum_{n=1}^{d} \langle E_k, \boldsymbol{U}_m \cdot \nabla E_n \rangle A_m A_n + \langle E_k, F_a \rangle.
\tag{A3}
$$

Given that $\{A_1, A_2, \ldots, A_d\}$ form an orthogonal basis, i.e., $\langle A_k, A_n \rangle = \delta_{kn} C_k$ for a given mode $k$ (with $\delta_{kn} = 1$ for $n = k$, and $\delta_{kn} = 0$ for $n \neq k$), with $C_k$ being the eigenvalue of mode $k$, the mode equation budget is computed as

$$\frac{dA_k}{dt} = \frac{1}{C_k} \left( -\sum_{n=1}^{d} L_{kn}^A A_n - \sum_{n=1}^{d} L_{kn}^B A_n - \sum_{m=1}^{d} \sum_{n=1}^{d} N_{kmn} A_m A_n + F_k \right), \tag{A4}$$

which is the expression of Eq. (4) in Section 2.2.

Here we show the justification for the choice of truncation at the 1000th EOF mode and the choice of low modes. The first 1000 modes together explain $\approx 100\%$ of the variance of the PV anomalies of all winter days in ERA-Interim. The powers of the first 25 EOF modes are concentrated in the period longer than one week. Therefore, we defined mode 1-25 as low modes.

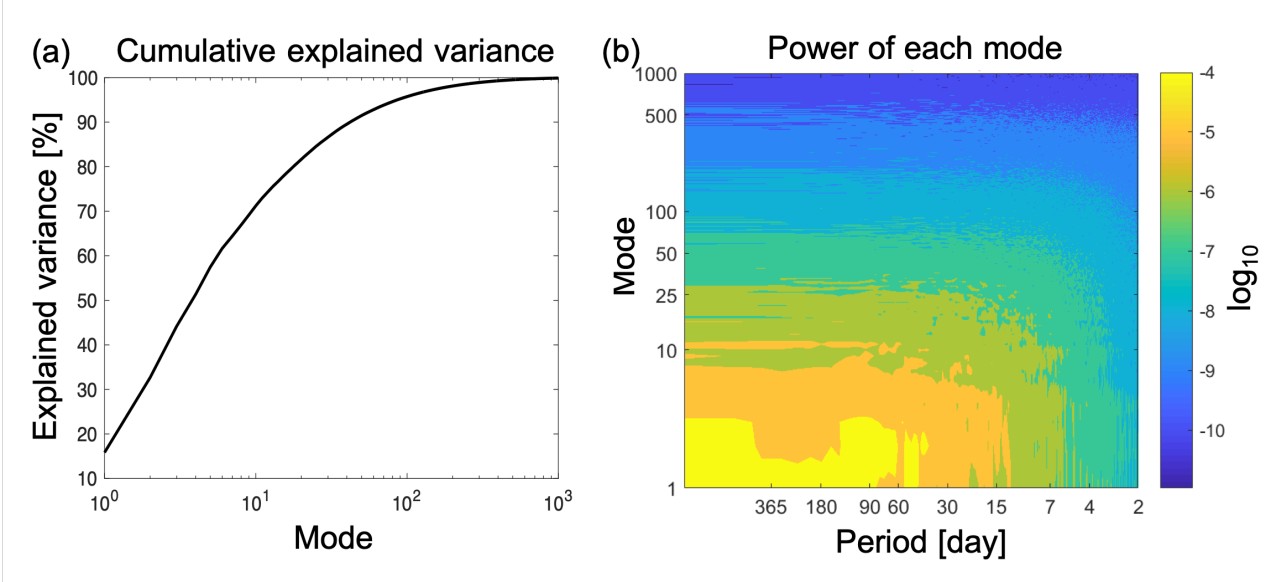

**Figure A1.** (a) The cumulative explained variance of the first 1000 EOF modes of PV. (b) The power spectrum of the first 1000 EOF modes.

**Appendix: B. Relation between the rate of change of $A_1$ and the PV flux**

In this appendix, we show the derivation for obtaining approximations of the linear and nonlinear terms (Eq. (9)) in the $A_1$
tendency equation using the PV flux form. Taking the inner product between Eq. (8) and $E_1$, and neglecting the variation of $\frac{1}{\rho_\theta}$

in the inner products, we obtain the following approximation of the rate of change of $A_1$:

$$C_1 \frac{dA_1}{dt} \approx -\frac{1}{\rho_\theta} \sum_{n=1}^{d} \langle E_1, \nabla \cdot (\rho_\theta E_n \boldsymbol{V}_c) \rangle A_n - \frac{1}{\rho_\theta} \sum_{n=1}^{d} \langle E_1, \nabla \cdot (\rho_\theta P_c \boldsymbol{U}_n) \rangle A_n -$$

$$\frac{1}{\rho_\theta} \sum_{m=1}^{d} \sum_{n=1}^{d} \langle E_1, \nabla \cdot (\rho_\theta E_n \boldsymbol{U}_m) \rangle A_m A_n - \langle E_1, P \frac{1}{\rho_\theta} \frac{\partial \rho_\theta}{\partial t} \rangle + \langle E_1, F_a \rangle, \tag{B1}$$

where $C_1$ is the eigenvalue associated with the first EOF mode of PV. Comparing the linear and nonlinear terms in Eq. (B1) with those in Eq. (A3), we can see that

$$L_{1n}^A + L_{1n}^B \approx \langle E_1, \nabla \cdot (\rho_\theta P_a \boldsymbol{V}_c) \rangle + \langle E_1, \nabla \cdot (\rho_\theta P_c \boldsymbol{V}_a) \rangle,$$

$$N_{1mn} \approx \langle E_1, \nabla \cdot (\rho_\theta P_a \boldsymbol{V}_a) \rangle. \tag{B2}$$

As we mentioned in Section 2.3, given the wavenumber-0 structure of $E_1$, we further approximate $E_1$ to be only a function of latitude. Thus, taking the inner product with $E_1$ can be approximated as taking a latitude-weighted integral of the meridional gradient of PV flux as demonstrated below:

$$\langle E_1, \nabla \cdot (\rho_\theta P_a \boldsymbol{V}_c) \rangle + \langle E_1, \nabla \cdot (\rho_\theta P_c \boldsymbol{V}_a) \rangle \approx 2\pi a^2 \int_{\phi_1}^{\phi_2} E_1(\phi) \frac{\partial [\rho_\theta P_a^{**} v_c^*] + [\rho_\theta P_c^{**} v_a^*]}{\partial y} \cos\phi \, d\phi,$$

$$\langle E_1, \nabla \cdot (\rho_\theta P_a \boldsymbol{V}_a) \rangle \approx 2\pi a^2 \int_{\phi_1}^{\phi_2} E_1(\phi) \frac{\partial [\rho_\theta P_a^{**} v_a^*]}{\partial y} \cos\phi \, d\phi, \tag{B3}$$

where $a$ is the radius of the Earth, and $\phi$ is the latitude with $\phi_1 = 30°\text{N}$ and $\phi_2 = 90°\text{N}$. Using Eq. (B3), we can obtain the approximated linear and nonlinear terms from Eq. (9).

## Appendix: C. EOF modes and PC time series of PV in the simplified Isca model

In this appendix, we show the spatial patterns of the first 10 EOF modes of PV anomalies at 850 K and the associated first PC time series in the Isca model as described in Section 2.1. The first 10 modes together explain $\approx 82\%$ of the variance of the PV anomalies of all winter days in Isca model.

## Appendix: D. Mode decomposition budget for displacement and split SSWs in ERA-interim

In this appendix, we show the $A_1$ mode decomposition budget for the composites of displacement and split SSW events in ERA-interim, which can be compared with Figure 3.

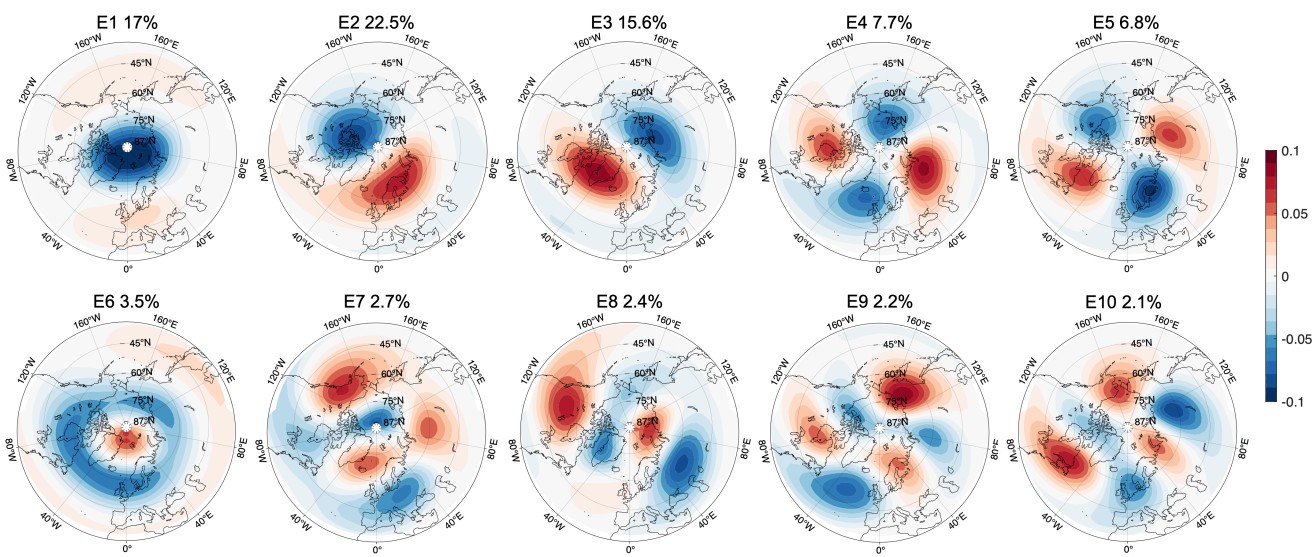

**Figure C1.** The first 10 EOF modes of PV at 850 $K$ ($E_1, E_2, \ldots, E_{10}$) of the combined set of basis vectors as described in Section 2.1 using the simplified Isca model daily data. The percentage number indicates the variance explained by each EOF.

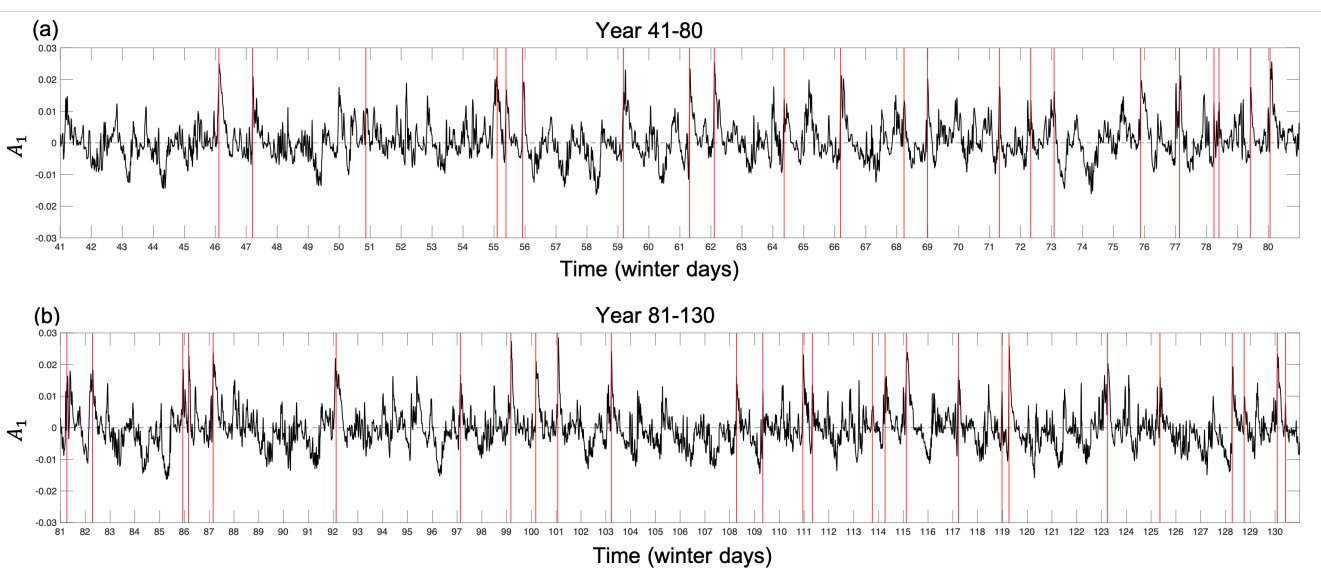

**Figure C2.** The PC time series ($A_1$) of winter days corresponding to the first EOF spatial pattern ($E_1$) using Isca-model winter daily data (from October to April) for (a) Years 41-80 and (b) Years 81-130. The PC time series for the winter days of Years 1-40 can be seen in Figure 2b. The red lines indicate the onset dates of SSW events.

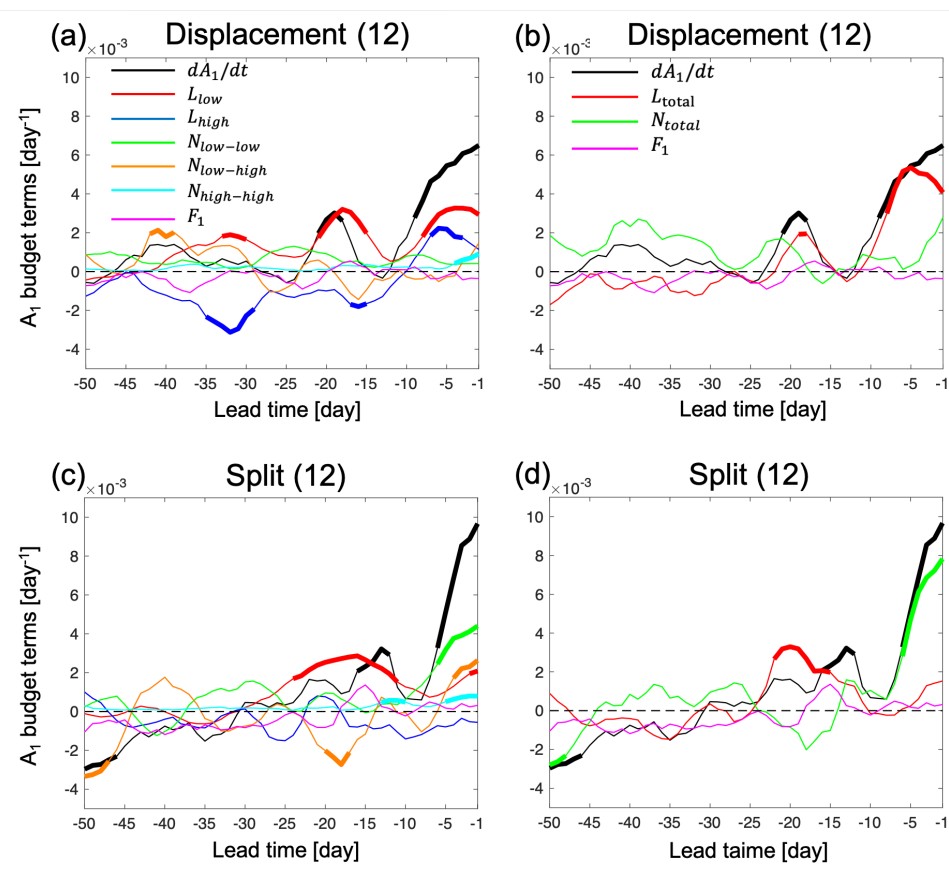

**Figure D1.** The composite of the $A_1$ budget as a function of lead time from 50 to 1 days before the onset of events in ERA-Interim. (a, b) composite of displacement events, (c, d) composite of split events. (a, c) show each term of the mode equation budget (Eq. (5)) separately, and (b, d) show the sum of the linear and nonlinear terms for the two types of events. The number in the bracket in each panel title indicates the number of SSW events. A 5-day running mean is applied to all lines. Bold lines indicate the values that are outside the 2.5th to 97.5th percentile range of normal winter days values from bootstrapping as described in the text. The representation of each line color in (c) and (d) is the same as the legend in (a) and (b), respectively.

*Author contributions.* Z.W. performed the derivations, data analysis, made the figures and wrote the first draft. B.J.-E. performed the Isca model computations. Z.W. and D.D. designed and supervised the study. All authors provided feedback for the manuscript and helped with discussions of the analysis.

*Competing interests.* The authors declare no competing interests.

*Acknowledgements.* The work of Z.W. and R.d.F. was funded by the Swiss Data Science Center within the project *EXPECT* (C18-08). Funding from the Swiss National Science Foundation to B.J.-E. and D.D. through project PP00P2_170523 is gratefully acknowledged. **This project has received funding from the European Research Council (ERC) under the European Union's Horizon 2020 research and innovation programme (grant agreement No. 847456).** The authors would like to thank two anonymous reviewers, John Albers, and the co-editor Thomas Birner for their helpful comments throughout the revision of the manuscript.

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
