# Peer review of "Emergence of representative signals for sudden stratospheric warmings beyond current predictable lead times"

_Weather and Climate Dynamics, 2021_

## Author Comment (AC1)

**Response to Reviewers**

We would like to thank John Albers and the two anonymous reviewers for careful reading, insightful comments and helpful suggestions for our study. These have been included into the manuscript (see changes indicated in **bold** in the annotated manuscript attached at the end of the reviewer's response). Please find below the detailed responses (in blue) to the reviewers' comments and suggestions. All line indications refer to the new (annotated) version of the manuscript.

The main changes to the manuscript are listed here:

1. We changed the title of the manuscript to "Emergence of representative signals for sudden stratospheric warmings beyond current predictable lead times".

2. We have reframed the implications of our study. We do not aim to improve the predictability of SSWs, and thus we keep the word "predictability" only for literature review. However, the increasing positive values of the tendency of the first PC time series at lead time of 3-4 weeks before the majority of SSWs suggest that the intrinsic predictability of SSWs may be longer than the current two-week predictability shown in the forecasting models. Our results can be viewed as a promising step towards improving the predictability of SSWs in the future by using more advanced statistical methods or in operational forecasting systems.

**Response to John Albers:**

This is an interesting paper that does a nice job investigating the relative role of linear and nonlinear processes for driving stratospheric variability, with important implications for subseasonal predictability. In particular, I think it is interesting that the results suggest that linear processes likely dominate downward propagating stratospheric anomalies until the final week or so prior to an SSW, at which point nonlinear processes seem to become important. To be clear, my comments below are not really criticisms of your results, rather I am just trying to mention a few issues that I think would be useful to keep in mind when interpreting what your results imply both in terms of physical process understanding and subseasonal predictability. (Note, all references that I cite are listed at the bottom of this review)

Thank you for your detailed comments and explanations. Below are point-by-point responses, and we also add some discussion in the revised manuscript in response to your comments.

To begin, I should point out that I think that the overall methodology of employing EOFs as a filter is, for the most part, totally fine. Indeed, in many respects, an 'EOF-filter' is a better

filter technique than a spectral filter because it implicitly takes into account both spatial and temporal information. That said, I think that a little bit of caution needs to be used when interpreting what a single EOF represents (here I am mostly, though not completely, referring to your 'SSW-EOF', $E_1$).

As nicely discussed in Monahan et al. (2009, in particular their Sect. 3), individual EOFs generally do not coincide with the dynamical/physical modes of the dynamical system from which they are derived. One reason for this, is that the atmosphere and ocean (and certainly the coupled atmospheric-oceanic system) contain processes with very different timescales, which results in a 'nonnormal' dynamical system. This concept (i.e., 'nonnormality') is relatively easy to understand in terms of something like the NAO, where distinct physical processes like ENSO, the MJO, synoptic-eddy feedbacks, and SSWs all project onto the NAO pattern. This means that total NAO variance (as represented by the 1st EOF of Atlantic MSLP or geopotential) is not a single physical/dynamical mode, but rather a convolution of variance that arises due to many individual physical processes/dynamical processes. That is, any given NAO anomaly is a result of constructive/destructive interference between the different types of variability (i.e., ENSO, MJO, etc.).

Thank you for your insight. We agree that the EOF modes cannot be interpreted by default as physical modes, as e.g. also discussed in Dommenget and Latif (2002), however they are often able to capture structures and dynamics that are intrinsic to the studied phenomena. In our case, the pattern captured by $E_1$ indicates that PV anomalies are centered at the polar cap. Because $E_1$ is derived using *only* days around the SSWs, $E_1$ is targeted towards SSWs by design and the variance explained by $E_1$ for the days around SSWs is around 28%, which is typically large compared to what can be seen in many climate studies looking at the first EOFs, i.e., often around 10-15%. Here the physical process (dynamics) represented by the variation of $E_1$ is clear, i.e., it represents the changes in the strength of the polar vortex, and it is strongly correlated with polar cap temperature. Since the weakening of the vortex is the dominant behavior of the polar vortex during SSWs, we believe that $E_1$ can be used to represent the weakening of the polar vortex even before the occurrence of SSWs, which is supported by the fact that the largest values, i.e., extremes, in the first PC (indicated by red vertical lines in Fig. 2) correspond to SSW events. We have added references and the discussion in Lines 149-152.

Now, thinking in terms of nonnormality has important implications for your results, because (1) it is important for interpreting what physical processes may give rise to your $E_1$ EOF, and (2) it will dictate what portion of the $E_1$ EOF variance might be predictable at various forecast leads. To help explain what I mean by this, below I use one possible scenario as an example (the PJO/NAM). To be clear, I do not know to what extent this scenario is applicable to your results (though I would guess it is), but the scenario itself is perhaps less important than thinking about what nonnormality means in terms of your $E_1$ EOF and the predictability of the variance that this EOF represents.

The potential scenario that I consider here is that your $E_1$ EOF represents a broader class of

stratospheric variability that regularly occurs and is occasionally punctuated by an SSW event. This is an idea envisioned by several previous authors (e.g., Kodera et al. 2000 and Kuroda and Kodera 2004) where it is postulated that the polar night jet oscillation (PJO) can be considered to be a general class of downward propagating stratospheric anomalies, that may occasionally be punctuated by a particularly strong PJO-like event in the form of a SSW. Because the PJO is typically identified via EOFs, it therefore likely arises from the nonnormal dynamics of ENSO, the MJO, the QBO, internal variability etc. Obviously some of these processes are more predictable at subseasonal leads than others. The SSW on the other hand, may typically (though not always) be due to internal nonlinear stratospheric dynamics (e.g., Sjoberg and Birner 2014, Birner and Albers 2017, White et a. 2019, de la Camara 2019, Nakamura et al. 2020), which are unlikely to be predictable beyond 1-2 weeks (i.e., variability governed by the deterministic limit of predictability).

So, what does this mean in terms of your results and subseasonal predictability? One possibility is that the portion of $E_1$ that you find behaves linearly is part of a broader class of PJO or NAM like variability that is not necessarily indicative of a future SSW, but may, under certain circumstances, be predictable on subseasonal timescales as you have suggested. On the other hand, the nonlinear portion of $E_1$ may necessarily be related to SSWs only, and may never be predictable beyond synoptic forecast leads (1-2 weeks). Complicating matters further, is the open question of whether PJO/NAM events without a SSW are strong enough to generate predictable anomalies in the troposphere.

We thank the reviewer for the detailed example and explanation. We agree that the linear component of the PV advection that we found for the lead times of around 3-4 weeks may not be specific to and uniquely representative of SSWs. The same is true for the increasing wave activity at lead times of around 3-4 weeks, which does not guarantee a future SSW either. However, both the linear component of the PV advection and the convergence of wave activity flux lead to a weakening of the polar vortex. When we look at the distribution of $\frac{dA_1}{dt}$ amongst all 25 SSW events in ERA-interim as shown in the figure below (Figure R1), we see that more than 60% of SSWs have positive values of $\frac{dA_1}{dt}$ at lead times of 20 days and the median of the distribution is consistently positive from 20 days onwards before the onset of the events. We agree with the reviewer's comments that this signal could be representative of a PJO event. However, it would be difficult to distinguish these signals, as in reanalysis, almost all PJO events are associated with a SSW event, while the reverse it not true. Hence, we think that while the detected signals may also pick up features of PJO events, they are certainly representative of SSW events. Furthermore, the fact that a large number of SSWs display a strong and persistent linear signal indicates the importance of the linear term in preconditioning/modulating the onset of SSWs. Even though the linear component may not be unique to SSWs, we found that large positive values of $\frac{dA_1}{dt}$ (larger than one standard deviation of $\frac{dA_1}{dt}$ during winter) correspond to either an SSW event or a strong polar vortex deceleration event (a deceleration of the zonal-mean zonal wind at 10 hPa and 60°N of more than 16.4 m/s in 10 days, which corresponds to the 60th percentile in terms of the strength of all wind deceleration events). $\frac{dA_1}{dt}$ and its

linear contributor for SSWs events tend to be stronger and more persistent than for the strong deceleration events. Figure R2 shows $\frac{dA_1}{dt}$ of the composite of the strong deceleration events (red) compared to that of the composite of SSW events (blue). Note that SSW events are a subset of the strong deceleration events and thus we exclude the SSW events from the strong deceleration events in the plot, such that the two sets of events are disjoint to avoid any overlap and ambiguity. The signals shown at lead times of around 20 days appear almost exclusively for SSWs. We added a discussion on this topic in Lines 556-566.

Concerning the nonlinear contributor of $\frac{dA_1}{dt}$, we found that it is only important for wave-2 SSW events and has almost no contribution for wave-1 SSWs. Therefore, we will use the nonlinear signals at lead times of 10 days to one week to infer the type of event.

Concerning the downward influence of the stratospheric anomalies in the troposphere, we did not look into whether PJO/NAM events without the reversal of the wind – note that there are very few of these events – can still have an influence on tropospheric weather and we actually did not discuss the downward impact of SSWs in the study. We think this is a very interesting and important question to investigate but it is beyond the scope of the current study. From another perspective, if the PJO/NAM events have a similar downward impact, then the signals found here could also be useful to capture these events.

[Figure]

Figure R1: The distribution of $\frac{dA_1}{dt}$ for all 25 SSW events in ERA-interim as a function of lead time. The bottom and top of each box are the 25th and 75th percentiles of the distribution. The red line in the middle of each box is the median of all SSW events. The whiskers extending above and below each box are the maximum and minimum. The red plus sign indicates the outliers, which lie at more than 1.5 times the interquartile range from the bottom or top of each box. The black line shows the mean of all SSW events and it is the same as the black line in Figure 3a and 3b. Bold parts of the black line indicate the values that are outside of the 2.5th to 97.5th percentile range of normal winter day values determined from bootstrapping as described in the main text.

[Figure]

Figure R2: Composites of $\frac{dA_1}{dt}$ as a function of lead time from 50 to 1 days before the onset of events. Blue is for SSW events and red is for strong deceleration events.

Overall, I think that it is probably important for you to comment in your paper on what can (or cannot) conclusively be physically implied about what your $E_1$ EOF represents. For example, what processes might give rise to the potentially predictable (linear) behavior that you have identified 25 days prior to SSW onset? Is the SSW a culmination of those linear processes that somehow transition to nonlinear behavior (for e.g., finite amplitude ideas such as Nakamura et al. 2020 or resonance of some kind)? Or does the nonlinear behavior occur independently from the linear processes? In addition, in the future, it would be useful to determine whether $E_1$-type variability that occurs with or without a SSW might imply different levels of enhanced tropospheric subseasonal skill. In other words, if an SSW is required in order to make the stratospheric anomaly large enough to be associated with enhanced tropospheric forecast skill, but the SSW is ultimately only predictable 1-2 weeks ahead of time, then does that mean that the weaker stratospheric anomalies that are linearly predictable at 3-4 week leads may unfortunately be of lesser practical importance for forecasting tropospheric anomalies if they occur without a SSW? On the other hand, if $E_1$-type variability can be used to guide tropospheric forecasts even without a SSW occurring, that would be very useful information as well.

We agree with the reviewer that the question of which processes lead to the increase of the linear and nonlinear contributions in the $A_1$ budget is important. In this study however, our focus is on the spatial patterns of linear and nonlinear PV flux terms as they can help us understand the linear and nonlinear contributions to the increase of $\frac{dA_1}{dt}$. We find that the spatial pattern

of the linear PV flux for wave-1 SSW events is largely stationary (Figure 7), which is due to the stationary spatial patterns of PV and $v$ anomalies (Figure 8). Understanding the question of why the PV and $v$ anomalies show these stationary patterns could help answer the question of which processes lead to the increase of linear and nonlinear contributions. However, in this study we do not focus on connecting the persistent anomalies to certain tropospheric weather events, like blocking, mid-latitude cyclones, etc. From the spatial pattern of the linear and nonlinear PV flux (Figure 7), the polar vortex tends to be displaced for wave-1 events and split for wave-2 events due to the rearrangement of the PV by the PV flux. As Reviewer 1 points out, our results are consistent with the results from Smith and Kushner (2012), and we cite the related previous studies that connect different processes to the linear and nonlinear contributions to the increase of $\frac{dA_1}{dt}$. For example, Smith and Kushner (2012) (and the references therein) suggested that the displacement events are preceded by sea level pressure anomalies associated with the Siberian high which is consistent with the increase of the linear meridional heat flux before the events. As another example, the increasing nonlinear signals from one week before the onset of wave-2 SSWs could be due to the resonance mechanism mentioned in Albers and Birner (2014). We agree that enhanced prediction skill in the troposphere is a very important question and worth further analysis. However, in the current study our focus is on finding dynamical circulation signals representative of SSWs. We leave the connection with the predictability of the troposphere for future work. We added a discussion on this topic in Lines 567-581.

Again, I don't have any conclusive answers to the above questions, but it is probably worth pointing out that your results do appear to generally agree with some recent results that myself and a co-author recently published (Albers and Newman ERL Feb. 2021). In short, our results suggest that linearly predictable stratospheric anomalies are associated with enhanced tropospheric predictive skill of the NAO. Of relevance here, is the fact that our results have some interesting similarities to what you have found in your paper. For example, similar to your results, we find that strong downward propagating stratospheric NAM anomalies are generally associated with linear processes for lags as far back as 25-30 days prior to 'stratospheric NAM event onset'. Likewise, we also find that nonlinear processes only become important 0 to 15 days prior to 'event onset' (denoted by stippling in our Fig. 1b). Interestingly, we were able to identify two types of dynamical modes (note, these are not EOFs), one single mode representing purely stratospheric processes (related to the NAM), and a second collection of modes representing coupled tropical tropospheric-stratospheric processes. In terms of subseasonal predictability, these modes account for a small fraction of overall NAO variance (see our Fig. 5c), which helps explain why subseasonal forecast skill is so low on average. We did not, however, provide any insight into which processes (purely stratospheric vs. tropical-stratospheric) are more important for subseasonal predictability (that is, we did not address the questions outlined in Domeisen et al. 2019 or Afargan-Gerstman and Domeisen 2020). On this note, does your $E_1$-EOF have any relationship to any forms of tropical variability?

Thank you for the insights and questions. While this study is not looking at tropospheric predictability specifically, we believe that the findings in your study mentioned in the comments

are very interesting and the question of which processes are the most relevant for subseasonal predictability is important. Even though we cannot provide an answer to the question, we combined the answer to the previous comment with this answer, and added the discussion in Lines 567-581.

In closing, you mention in your paper that you would like to explore the implications of your work in the context of actual subseasonal predictability. Given that you already have identified an EOF that you believe is important, one quick test you could do would be to project IFS hindcast data (or whatever your preferred S2S model is) onto your $E_1$-EOF and then calculate 'forecasts of opportunity' as periods when the forecasted $E_1$-EOF loading is particularly high. This would a very rough way of identifying when the 'signal' part of a signal to noise calculation was particularly high, which typically equates to periods of higher forecast skill. If your $E_1$-EOF is identifying a 'skillful' portion of stratospheric variance, then these high loading periods may be associated with higher tropospheric skill (this type of 'signal' calculation is not as complete as computing the actual 'signal-to-noise ratio' as we did in our ERL paper, but it is easy to compute and seems to work reasonably well some circumstances, e.g., Albers et al. WCD 2021).

Thank you for your suggestions. In fact, we have already projected ECMWF hindcast data onto E1 extracted from ERA-interim and found that more than 70% of SSWs events can be identified using $\frac{dA_1}{dt}$. However, these results are part of ongoing work and will be the subject of a future manuscript. One major reason for not including these results in the current manuscript is that the model output from the S2S models does not contain PV, hence we had to re-write our equations using different variables, which required additional deliberations and care with regard to the analysis and results.

One final minor comment... for the de la Camara et al. paper that you reference, the 'de' and the 'la' are not capitalized.

Thank you for catching this. Change made.

**Response to Reviewer 1:**

My first comment refers to the predictability of SSWs by analyzing the tendency of the PC of the first variability mode of PV. I was wondering if the authors have also studied the number of events that present a rapid increase of this PC but they are not SSWs. How many false alarms would you get? I guess these "false alarms" would correspond to minor stratospheric warmings. I am also curious about the opposite case. Have the authors identified any SSW that would not

be associated with a fast increase of this PC? In this regard, I am a bit concern about the first assumption of linearity to identify SSWs (i.e. the use of a PCA to identify these events). My concern is based on the importance of nonlinear processes in the development of some of these events (wave- 2 SSWs) that is shown later in the study.

Thank you for your comments. The goal of the current manuscript is not to use $\frac{dA_1}{dt}$ or the corresponding contributors to identify or predict individual events but to find stratospheric signals representative of SSWs (from composites). For example, we found that $\frac{dA_1}{dt}$ emerges at lead times of 3-4 weeks, which might indicate that the dynamical predictability of SSWs could be longer than 1-2 weeks, as has been commonly observed in the literature. The issue raised by the reviewer is an important question to address, and this is the topic of our ongoing study. If we look at the distribution of $\frac{dA_1}{dt}$ for all 25 SSW events in ERA-Interim as shown in Figure R1 (see above, in the previous response to John Albers), one can see that more than 60% of SSWs have positive values of $\frac{dA_1}{dt}$ at lead times of 20 days. The median of the SSW events (red line within the box) consistently shows positive values of $\frac{dA_1}{dt}$ from around 20 days before the onset of the events. On the other hand, Figure 2a in the manuscript shows that there are some peaks in $A_1$ that do not correspond to an SSW event. However we found that all the peaks (values larger than 0.01, which corresponds to one standard deviation in the $A_1$ winter time series) correspond to a strong deceleration event (the deceleration of the zonal-mean zonal wind at 10 hPa and 60° N by more than 16.4 m/s in 10 days, which corresponds to the 60th percentile in terms of the strength of all wind deceleration events), which includes all SSW events. Since this study focuses on SSWs, we did not further investigate how different components in the PV equation contribute to $\frac{dA_1}{dt}$ in the strong deceleration events and how they are different from SSW events. In ongoing work, we are using a time series classification-based method to study these differences and apply S2S re-forecast data to look into the accuracy (e.g., hit rate and false alarm rate) of using $\frac{dA_1}{dt}$ at different lead times to identify an SSW event. We added a more careful discussion concerning these points in the conclusion in Lines 556-566.

Secondly, I am not sure if the authors are aware of the study by Smith and Kushner (2012) where they analyze the evolution of anomalies of eddy heat flux and their contributing terms at different vertical levels for displacement and split SSWs. Their results highly agree with some of this manuscript, particularly the relative role of linear and nonlinear processes and their timing for the development of displacement and split SSWs. It is true that there is not a straight relation between split and displacement SSWs and wave-2 and wave-1 events, but Ayarzaguena et al. (2019) repeated the analysis for the latter and found similar results. I think that it would be interesting to discuss and compare the results of the present manuscript with those of Smith and Kushner (2012).

Thank you for mentioning this study. As the reviewer points out, their results are highly consistent with our results. In fact, as we show in section 4 related to the physical interpretation of the mode decomposition, we can use the PV flux (and thus the wave activity) to interpret the results shown in section 3. To be more specific, we found that the linear (nonlinear) PV

flux is more important for the wave-1 (wave-2) SSWs, which agrees with the results on vertical wave activity in Smith and Kushner (2012). On the other hand, we also found that the linear and nonlinear terms are related more to wave-1 and wave-2 events than displacement and split events. We included a discussion in lines 323-327 and lines 417-421.

Specific comments

Lines 340-343: I was wondering if it would be a good idea to analyze SSWs in long simulations of more complex models such as CCMs. For instance, some CMIP6 models include interactive chemistry and provide daily output of long piControl simulations. I am suggesting this because I had the impression that the conditions in the ISCA model simulations are different from those in the reanalysis. For instance, there is no interannual variability in SSTs, but a strong warming in the equatorial Pacific is imposed. The evolution of the tendency of the PC1 is also different in the reanalysis and the model. It seems that SSWs might be only predicted in advance much later in the model than in reanalysis. Thus, it is not clear if the difference between reanalysis and model results is due to the short reanalysis sample or model biases.

Thank you for this comment. There are several potential reasons for why the behaviour of $\frac{dA_1}{dt}$ and its contributors in the Isca model is not the same as that in the reanalysis. We agree that model biases and lack of representation of certain processes in the Isca model are part of the explanation for the differences with the reanalysis. We think that using the CMIP6 models with long piControl simulations as comparison is a good idea. However, the output daily data of CMIP6 only has 8 pressure levels, which is not enough to yield an accurate computation of potential vorticity and its budget. For this reason of lack of sufficient data, we did not use CMIP6 models for comparison. To better understand the influence of the number of events and the model biases on $\frac{dA_1}{dt}$ in the Isca model, we randomly select 25 events from the 78 SSWs in the Isca model run and show $\frac{dA_1}{dt}$ in blue in figure R3. The clear increase of $\frac{dA_1}{dt}$ occurs around 10 days before the onset of the events, same as that of all SSWs in the Isca-model (red). Additionally, we used the output data to compute the $A_1$ budget from a more complex climate model (ICON, Zängl et al. (2015), Giorgetta et al. (2018)) with prescribed observed SSTs, $CO_2$, ozone, and aerosols concentrations following the monthly mean evolution of the 1979-2015 historical period (which contains most of the ERA-Interim period used in this study). The prescribed monthly mean prescribed values of SST allow the model to represent interannual variability and the teleconnections associated with El Niño-Southern Oscillation (ENSO). The ICON model has a higher resolution (∼158km horizontally) and uses parameterizations of gravity wave drag and cloud microphysics, processes that are not included in the Isca model. There is a total of 42 SSWs in the ICON simulation. $\frac{dA_1}{dt}$ of the ICON model simulation (green) shown in figure R3 and its variation is similar to that of the Isca model (red and blue) but with a clear increase at around 15 days before the events. If we randomly select 25 out of the 42 SSWs in ICON and repeat the analysis, the result is very similar (magenta). The figure suggests that for more complex models, the increase of the first PC tendency starts earlier and the tendency is more similar to the reanalysis (black). However, the large values of the tendency at around 20 days

cannot be reproduced in both the simplified and complex models used in this analysis and seem to be more related to model biases and variability than the averages of a large number of SSWs in the models. We modified the sentences in Lines 374-378 to clarify this aspect.

[Figure]

Figure R3: $\frac{dA_1}{dt}$ as a function of lead time from 50 to 1 days before the onset of SSW events in different data sets. Black is for the 25 SSWs in ERA-Interim; red is for the 78 SSWs in Isca model output; blue is for 25 SSWs randomly selected from the 78 SSWs in the Isca model; green is for the 42 SSWs in ICON model output; pink is for 25 SSWs randomly selected from the 42 SSWs in the ICON model.

Lines 378-385: I would also highlight the positive (and statistically significant) values of the linear term for wave-2 SSWs just before the onset of events. Their values are of opposite sign to those for wave-1 events.

We modified the sentences in Lines 421-423 to read: "The linear zonal-mean PV flux for wave-2 SSWs is positive and statistically significantly different from the normal winter days just before the onset of SSWs, while the linear zonal-mean PV is negative and also statistically significantly different from normal winter days for wave-1 SSWs."

Lines 426-445: I wonder if it would be helpful to represent in the same plot $v_a^*$ and $P_a^{**}$, one in contours and the other in shading. This might help to clarify the relation between $P_a^{**}$ and $v_a^*$ and nonlinear PV flux.

We now changed Figure 9 and plotted $v_a^*$ (green lines) on top of $P_a^{**}$ (shading). Note that even though the main features of nonlinear PV flux as shown in Figure 7 can be roughly inferred by $v_a^*$ and $P_a^{**}$ in Figure 9, some of the weak features cannot be represented well. This is because

$\{P_a^{**}v_a^*\} \neq \{P_a^{**}\}\{v_a^*\}$, where $\{\}$ denotes the composite mean of SSW events. We added a discussion on this topic in Lines 483-485.

Technical corrections:

Line 348: I could not find the results that the authors mention in Figure 3 of Ayarzagüena et al (2019). Are the authors maybe referring to Figure 3 of Ayarzagüena et al (2018)?

Thank you for spotting this. Change made.

Line 382: I think a "be" before "found" is missing.

Thanks, change made.

**Response to Reviewer 2:**

The mode decomposition analysis is a rather involved and difficult to interpret approach, but the authors have made significant efforts to connect their analysis to PV flux and wave- mean flow interaction diagnostics which are helpful. The results regarding the distinct processes leading to a wave-1 and a wave-2 warming are broadly consistent with other works that argue that wave-2 events evolve more rapidly, more non-linearly, more barotropically, and less predictably than wave-1 events.

However, much is made of the statistically significant precursors found in the time series of the leading EOF, their main metric for describing the evolution of the vortex; it motivates the title of the paper. Unfortunately, statistically significant composite signals preceding events don't imply anything about the predictability of the event in question. Just because the vortex weakens somewhat on average prior to the event does not imply that every time the vortex weakens a sudden warming will follow 25 days later. In other words - the composite suggests at most that this is a necessary condition, not at all that it is sufficient. To demonstrate evidence of predictability, the authors would need to identify relevant precursor events without reference to the warmings themselves, then show that warmings are more likely to occur following that event. This issue comes up in discussion of tropospheric 'precursors' to sudden stratospheric warmings all the time - there are statistically significant signals in the composite evolution prior to sudden warmings, but they turn out to be relatively useless as predictors because only a small subset of the tropospheric events are followed by a warming.

We thank the reviewer for the comments. Overall we agree with the reviewer and reframe the

implications of our study. We only use the word "predictability" when necessary.

In this manuscript we only aim to find signals that are representative of SSWs during their dynamical development rather than improving the predictability of SSWs. We agree that the positive $\frac{dA_1}{dt}$ from the composite does not imply that $\frac{dA_1}{dt}$ can be used to predict individual SSW events. However, the positive $\frac{dA_1}{dt}$ in the composite is indeed a representative overall feature of SSWs. Based on the bootstrapping, $\frac{dA_1}{dt}$ shown in the composite at lead times of 3-4 weeks is positive ahead of most SSWs, not just for specific SSW events. More specifically, the distribution of $\frac{dA_1}{dt}$ for all 25 SSW events in ERA-interim (Figure R1) shows that the median of $\frac{dA_1}{dt}$ is consistently above zero and around 70% of SSWs have positive values of $\frac{dA_1}{dt}$ starting at lead times of around 20 days. We see the results of the present manuscript as a promising step that it is possible to improve the predictability of SSWs using more advanced methods. We also notice that large positive values of $\frac{dA_1}{dt}$ (larger than one standard deviation of $\frac{dA_1}{dt}$ of all winter days) correspond either to an SSW event or a strong polar vortex deceleration event (the deceleration of the zonal-mean zonal wind at 10 hPa and 60°N by more than 16.4 m/s in 10 days, which corresponds to the 60th percentile in terms of the strength of all wind deceleration events). However, the magnitude and persistence of the positive $\frac{dA_1}{dt}$ in strong deceleration events composite are not as strong as those in SSW events composite (see Figure R2).

Related to the comment on the precursor signals in the troposphere, we believe that the signals found here are more directly related to and representative of SSWs than the tropospheric signals (as the reviewer pointed out, the signals found in the troposphere which many studies phrased them as "precursors" are not always followed by an SSW event) from a dynamical point of view. The tropospheric signals may not lead to a deceleration of the stratospheric zonal wind or an SSW event as Albers and Birner (2017) showed that only 1/3 of SSWs can be related to tropospheric precursors. While tropospheric anomalies can help to enhance the wave activity and generate more planetary waves propagating into higher levels, many previous studies showed that the stratospheric basic state is crucial for the upward propagation of planetary waves into the mid-to-upper stratosphere (e.g., Jucker 2016; Albers and Birner 2017). Therefore, an SSW event is the product of the interaction between the planetary wave and the mean wind flow in the stratosphere. The main reason why we believe the positive $\frac{dA_1}{dt}$ is more directly related to and representative of SSWs is because $E_1$ represents the variation of the polar vortex by design and the linear and nonlinear PV advection terms in the PV equation contribute to $\frac{dA_1}{dt}$. The linear and nonlinear PV advection terms can be related to the PV flux form as we showed in session 4 and they represent the interaction between the mean flow and the planetary wave activity. In a word, $\frac{dA_1}{dt}$ represents not just one component (i.e., tropospheric signals or stratospheric mean state) but combined effect of both components. We added a discussion on this topic in Lines 556-566, including the following statement: "What we found here suggests that the intrinsic predictability of SSWs may be longer than the current two-week practical predictability. However, further work is still needed to investigate whether the predictability of SSWs can actually be extended, and if yes, how."

I have a few other comments, some more substantial than others, but addressing this concern is essential, and will amount to major revisions. The authors should either reframe the results from a dynamical point of view and remove reference to predictability, or show evidence for predictability that justifies this emphasis.

As we mentioned in the previous response, in this manuscript we only aim to find signals that are representative of the dynamical development of SSWs rather than improving the predictability of SSWs. Using the signals (e.g., the positive $\frac{dA_1}{dt}$) found in this study to identify each individual SSW event will be the subject of upcoming work. To clarify this, we modified the wording throughout the manuscript and reframed the implications of our results without addressing "predictability" (see the tracked changes version of the revised manuscript). We also changed the title of the manuscript "Emergence of representative signals for sudden stratospheric warmings beyond current predictable lead times".

**Further Comments**

Climatology

It's not completely clear from the methodology just how the climatology has been computed, but the authors should consider imposing some kind of low-pass smoothing filter on the climatology if they haven't done so already. Given the finite number of years in the calculation, particularly for observations, there will be considerable residual high- frequency sampling variability that can artificially increase the small scale variance in the anomalies. This probably won't impact the low frequency modes too much, but it will affect the details of the high-frequency modes.

The climatology here is the mean value of each day of all the years available. We added a comment in Line 130 to make the definition of climatology clearer: " ... obtained by computing the daily mean values of PV over all available years". As suggested by the reviewer, we applied a running-average (30-day running mean) to low-pass filter daily mean values over the available years. The results are shown in the figure below. The results are very similar to those shown in Figure 3. The main obvious difference is in $N_{low-high}$, which captures the interaction between the low and high frequency EOF modes without influencing our main conclusions. One reason for the very similar results between using low-pass filtered climatology and no low-pass filtered climatology is that applying PCA on the PV data already acts as a filter. The first PCs tend to capture low-frequency signals, while high-frequency signals (captured by further PCs) are discarded in the analysis. Our results show that the main contributions to the increase of $\frac{dA_1}{dt}$ come from the first 25 EOF modes. We added a sentence in Lines 131-135 to clarify this aspect.

EOF calculation

From what I could understand, the EOF calculation has been carried out in two steps. The first analysis is used to obtain the structure of the leading EOF, then this variability is removed from the PV field and a separate calculation is carried out to obtain the remaining modes. It's not clear why this two step approach is adopted. I think it's connected as well to the fact that

[Figure]

Figure R4: Same as Figure 3 in the manuscript but applying 30-day running mean of the daily climatology.

EOF 2 explains more variance than EOF 1 (Figs. 1, C1); I'm assuming that the percentage reported for EOF 2 is the fraction of the *remainin* variance that is explained by EOF 2. If this multi-step approach is important, this should be clearly explained and demonstrated.

The reason for using a two-step approach to obtain the EOF modes is that we want to obtain an EOF mode that best describes the variability of the polar vortex that is most directly related to the development of SSWs. This is achieved by computing the first EOF mode based only on data around the onset day of SSWs. Specifically, days -10 to +5 are used for this purpose. Applying PCA to all winter days leads to a first EOF mode that is different from the current EOF mode shown in Figure 1. The variance explained by EOF1 is smaller than that explained by EOF2 because EOF1 is derived only from data around SSW events. EOF1 is therefore concentrated on SSW characteristics, and it does not capture all the information contained in the whole winter data, which generally displays less variability than is observed around SSW events. The remaining EOFs are derived directly from the whole winter data after removing its projection onto EOF1. It should therefore not be surprising that EOF2 captures more variance of the whole winter data than EOF1. After we combined the EOFs from the two EOF mode decompositions, we computed the explained variance of this combined basis for the whole winter data. We added a discussion and clarification of this method in Lines 137-143 and Line 156-159.

Description of ISCA model

The processes and parameterizations used should be briefly discussed; calling it 'intermediate' complexity is a bit ambiguous. E.g. is there a gravity wave drag parameterization? Realistic radiation? Ozone variability? etc.

We now added a more detailed description of the model in Lines 107-115.

Bootstrap methodology

The bootstrap methodology for computing statistical significance is reasonable, but the authors should use non-SSW days that have the same seasonal distribution as the SSW days to avoid aliasing with the seasonal cycle.

Since we removed the daily climatology of PV to obtain the EOF modes and corresponding PC time series, the evolutions of the first PC and its contributors do not contain the seasonal cycle. As the reviewer suggested that the bootstrapping should reflect the temporal distribution of SSWs, we repeated the analysis to use non-SSW days only during December, January, February, and March (during the typical months of occurrence of SSWs). The results are very similar and do not change our main conclusions. We added a sentence in Line 276-279.

Minor comments:

-Sign convention for leading EOF: At present a positive A1 anomaly corresponds to a weakening of the vortex. This makes the descriptions awkward to me, e.g. in the abstract: the leading PC is an 'indicator of the strength of the polar vortex' but an increase indicates a deceleration. This took me a while to understand. The opposite sign convention might be more intuitive.

The first PC is an indicator of the strength of the vortex which can go in both directions, and positive PC values here indicate a weakening of the vortex, and therefore the potential

occurrence of SSWs. We added a footnote in the manuscript to clarify it.

l48: the statistics are closer to two in three winters.

Thanks, change made.

Part 1 of the two papers published by Domeisen et al. in 2020 is labeled Domeisen et al. 2020b, and part 2 is labeled Domeisen et al. 2020a. The citations are appropriate as labeled, but they confused me because I am used to the more sequential ordering.

This is unfortunately not something we can change in the latex template. The journal might be able to change this during copy-editing.

[revised manuscript text omitted]

---

## Author Response (AR2)

**Response to Reviewers**

We would like to thank the two anonymous reviewers for careful reading, insightful comments and helpful suggestions for our study in the second round of review. These have been included into the manuscript (see changes indicated in **bold** in the annotated manuscript attached at the end of the reviewer's response). Please find below the detailed responses (in blue) to the reviewers' comments and suggestions. All line indications refer to the new (annotated) version of the manuscript.

**Response to Reviewer 1:**

The authors have addressed carefully the reviewers' comments. The new explanations have improved the manuscript. I am particularly happy with the change in the title and the description of the implications of the study. In the previous version, I had some concerns about that.

I have only a couple of typos that I indicate below: L278: I would remove "a".

Change made

L354: leads → lead

Change made

L533: underlyingx → underlying

Change made

L419-423: The two sentences are somehow repeatitive in some terms. I would reformulate them. Thank you for the careful reading. Now we changed the sentence as:

Line 419-422: "Different from Smith and Kushner (2012), the nonlinear (linear) component of the PV flux even becomes positive just before the onset of wave-1 (wave-2) events (the positive linear PV flux in wave-2 is statistically different from normal winter days), counteracting the weakening of the polar vortex."

**Response to Reviewer 2:**

This is my second review of this paper. Thank you to the authors for their edits; these have more or less addressed my main concerns with the previous version of this paper. The discussion around predictability is generally more careful, though there are still statements in the conclusions regarding predictability that are not justified by the present analysis (lines 509, 514, 518). These should be qualified or removed. I appreciate the new paragraph discussing the issue of inferring predictability, but it still doesn't address the main logical gap in inferring predictability from the composite signals, which is that there is no evidence that the precursor signals (i.e. the enhanced linear and non-linear contributions to the tendency of the 1st PC time series) actually provide predictive information about the future occurrence of an increase in the PC 1 time series.

Thank you for your concern. We agree that predictability is not explicitly shown, and that we did not investigate predictability in terms of e.g. false positives. Our argument is based on the fact that most of the SSW events indeed show increasing values of linear PV advection and $\frac{dA_1}{dt}$. We thus suggest that our findings may hint at a potential for predictability, and we have changed the conclusion accordingly. The sentence about predictability on line 509 has been deleted.

I still feel there are many changes that could be made to improve the readability and impact of the work, but as it stands I don't have further major concerns with the methodology. In my opinion the manuscript would be acceptable after minor revisions.

Thank you for the comments. We made additional changes according to the reviewer's comments throughout the manuscript that we hope will clarify things further.

**Specific Comments**

*SSW classification*

The authors are at times fairly careful about distinguishing 'split' and 'displacement' classification from wave number one vs. wavenumber two classifications of warming events. It is clear these two ways of classifying events are pretty distinct from one another, even if they have some common features (e.g. discussion at end of p 8). This distinction should be reiterated in the conclusions.

We added a sentence in Lines 521-523: "The above differences are also present in the displacement and split SSW events, but the differences are somewhat smaller than those between wave-1 and wave-2 events, as not all split events are induced by wave-2 planetary waves."

*Model Description*

The model description is much improved. The vertical and horizontal resolution should be given in the first paragraph of section 2.1 (the horizontal resolution is stated later and the vertical resolution is not given). The total number of wintertime days for both the reanalysis and the

model should also be stated in the data description.

We added a sentence in the first paragraph of section 2.1 in Line 109-110: "it uses a T42 horizontal resolution and 50 vertical levels up to 0.02 hPa with 25 levels above 200 hPa."

We also added the total number of winter days for both reanalysis and the model.

*Mode decomposition*

The methodology is clearer now - thank you again to the authors. That being said, the whole approach is extremely complicated, with approximations being made at multiple stages, including the EOF decomposition and the transition from an advection to a flux framework. I'm not at all convinced that this complexity is needed, and it's certainly true that the work would have more impact if the methodology was more transparent. But I leave it to the authors to decide whether they wish to simplify things.

We thank the reviewer for the concerns. We have now clarified in the manuscript where approximations are made. In the first step, i.e. the mode decomposition, only a minimal approximation is made as the first 1000 modes that are used contain 99.99% of the variance. An approximation is only made in the interpretation of the results of the mode decomposition, e.g., where we used PV flux. We note that we did not apply mode decomposition to PV flux form as shown in Equ.(9). In order to clarify the use of approximations, we have amended the writing for the analysis and interpretation in the methodology section. The mode decomposition method uses spatial modes and PC time series to represent each term in the PV equation without any approximation, which is suitable to study the variations of the important mode that represent the change of the polar vortex.

A few questions though: how was it determined that 1000 modes were needed (l 175), particularly when later the number of modes is further truncated to 25 (l 205)?

We truncated at different numbers of EOF modes to investigate how many EOF modes should be included (see also answer to previous reviewer question). The analytical $\frac{dA_1}{dt}$ derived from Eq.(5) is equal to the actual $\frac{dA_1}{dt}$ computed from the central differences between $dA_1$ when using all EOF modes. As the number of EOF modes used increases, the estimation of $\frac{dA_1}{dt}$ becomes more accurate. Here, we find that when using 1000 EOF modes, the difference between the analytical $\frac{dA_1}{dt}$ and the actual $\frac{dA_1}{dt}$ is almost zero. If we use fewer modes, the differences between the analytical $\frac{dA_1}{dt}$ and the actual $\frac{dA_1}{dt}$ increases as expected, but the main results do not change as long as we use a sufficient number of EOF modes that can explain at least 90% of the variance of the fields. Figure r1(a) shows that the total explained variance from the first 1000 modes is very close to 100%, which is another justification for us to use 1000 modes. We use 25 modes to represent low-frequency EOF modes but we use the full 1000 modes to reconstruct the PV and wind fields. Figure r1(b) shows the power spectrum of the first 1000 EOF modes. For the first 25 modes, the power is concentrated in the period longer than one week. Therefore, we defined mode 1-25 as low modes. We added Figure r1 to the appendix of the manuscript.

[Figure]

Figure r1: (a) The cumulative explained variance of the first 1000 EOF modes of PV. (b) The power spectrum of the first 1000 EOF modes.

And the methodology distinguishes between linear terms arising from anomalous advection of climatological PV fields and those from climatological advection of anomalous PV fields. Yet this decomposition is not used, while the 'high' vs. 'low' decomposition is introduced. Did the authors look at the separate linear terms?

The main reason why the linear and nonlinear framework is introduced is because we found that the evolution of the spatial patterns of the linear and nonlinear PV advection (or PV flux) are distinct before the onset of SSWs as shown in Figure 7. We found that there is a strong cancellation between the two linear terms from anomalous advection of climatological PV fields and from climatological advection of anomalous PV fields in normal winter days. When approaching the onset of SSWs, these two linear terms are no longer balanced and lead to a strongly negative PV advection. The combination of these two linear terms results in the PV anomalies before SSWs. Thus, we did not separate these linear terms. Eq (4) shows only the step of how we get the final mode equation budget for the analysis as shown in Eq.(5). As shown in Eq.(5), we combined these two linear terms to a total linear term. We separate this total linear term into low- and high-modes contributions. That is why in the results section (Figure 3-5), we only discussed linear terms from the low- and high-modes contribution.

Finally, the authors point out that the correlative structures in the wind fields (near l 180) need not be orthogonal. It seems that neither is there any guarantee that they are complete; in principle there could be important wind structures that are not captured by this mode decomposition. What guarantee do we have that the decomposition is sufficient?

We understand the reviewer's concern. To clarify, the wind modes used in this study explain 99% of the total variance of the wind fields. The modes of the wind fields are computed such that they fulfill the following relation:

$$\boldsymbol{V}_a = \sum_{n=1}^{d} U_n A_n, \qquad\qquad 5$$

where $\boldsymbol{V}_a$ are the daily anomalies of the wind fields, $U_n$ are the spatial modes of $\boldsymbol{V}_a$, and $A_n$ are the PCs. Given the first 1000 modes have already explained 99.99% of the total variance of the field, the modes for the wind fields are almost complete and all the important wind structures are captured. We used these 1000 modes of the wind fields and PCs to reconstruct the wind fields and they are almost identical with the actual wind fields. Given that our study focused on the large scale perturbation, we believe that it warrants our use of mode truncation.

[revised manuscript text omitted]

---

## Author Response (AR3)

Response to co-editor

We appreciate the editor's comment, however we would like to keep the terminology "predictable lead time", as it is widely used in the predictability literature to mean the lead times on which an event can be predicted (Tiedje et al., 2012; Orth and Seneviratne 2013; Baehr and Piontek, 2014; Ding et al., 2018; Anna Borovikov et al., 2019).

We added an explanation to this terminology in the manuscript: Line 96: "the current predictable lead time (meaning the lead time on which an event can be predicted) of two weeks". We also added the acknowledgements to the reviewers and the editor. Thank you for your assistant during the revision.

References

Tiedje, B., Köhl, A., & Baehr, J. (2012). Potential predictability of the North Atlantic heat transport based on an oceanic state estimate. *Journal of climate*, *25*(24), 8475-8486.

Orth, R., & Seneviratne, S. I. (2013). Predictability of soil moisture and streamflow on subseasonal timescales: A case study. *Journal of Geophysical Research: Atmospheres*, *118*(19), 10-963.

Baehr, J., & Piontek, R. (2014). Ensemble initialization of the oceanic component of a coupled model through bred vectors at seasonal-to-interannual timescales. *Geoscientific Model Development*, *7*(1), 453-461.

Ding, H., Newman, M., Alexander, M. A., & Wittenberg, A. T. (2018). Skillful climate forecasts of the tropical Indo-Pacific Ocean using model-analogs. *Journal of Climate*, *31*(14), 5437-5459.

Borovikov, A., Cullather, R., Kovach, R., Marshak, J., Vernieres, G., Vikhliaev, Y., ... & Li, Z. (2019). GEOS-5 seasonal forecast system. *Climate Dynamics*, *53*(12), 7335-7361.